# Phase regulation enabling dense polymer-based composite electrolytes for solid-state lithium metal batteries

Qian Wu[1,2], Mandi Fang[3], Shizhe Jiao[4], Siyuan Li[1,2], Shichao Zhang[1,2], Zeyu Shen[1,2], Shulan Mao[1,2], Jiale Mao[1,2], Jiahui Zhang[1,2], Yuanzhong Tan[5], Kang Shen[5], Jiaxing Lv[5], Wei Hu[4], Yi He[3,6] & Yingying Lu[1,2] ✉

Solid polymer electrolytes with large-scale processability and interfacial compatibility are promising candidates for solid-state lithium metal batteries. Among various systems, poly(vinylidene fluoride)-based polymer electrolytes with residual solvent are appealing for room-temperature battery operations. However, their porous structure and limited ionic conductivity hinder practical application. Herein, we propose a phase regulation strategy to disrupt the symmetry of poly(vinylidene fluoride) chains and obtain the dense composite electrolyte through the incorporation of $MoSe_2$ sheets. The electrolyte with high dielectric constant can optimize the solvation structures to achieve high ionic conductivity and low activation energy. The in-situ reactions between $MoSe_2$ and Li metal generate $Li_2Se$ fast conductor in solid electrolyte interphase, which improves the Coulombic efficiency and interfacial kinetics. The solid-state Li||Li cells achieve robust cycling at 1 mA cm$^{-2}$, and the Li|| $LiNi_{0.8}Co_{0.1}Mn_{0.1}O_2$ full cells show practical performance at high rate (3C), high loading (2.6 mAh cm$^{-2}$) and in pouch cell.

Driven by the ever-increasing demands for high-energy-density, long cycle life and safe devices, developing advanced electrolytes compatible with lithium (Li) metal anodes and high-voltage cathodes (i.e., $LiNi_{0.8}Co_{0.1}Mn_{0.1}O_2$, NCM811) has been highly pursued over the past decades[1–3]. However, conventional organic liquid electrolytes (LEs) are plagued by flammability, Li dendrite growth and uncontrollable side reaction issues. Replacing LEs with intrinsically safe solid-state electrolytes (SSEs) is significant for developing safe and stable Li metal batteries (LMBs)[4–7], because the mechanical properties and electrochemical stability of SSEs can suppress Li dendrite growth and mitigate interfacial reactions to some extent. Among SSEs, solid polymer electrolytes (SPEs) are accepted as promising candidates due to their flexibility and interfacial compatibility[8–10]. Specifically, compared with other polymer-based SSEs, poly(vinylidene fluoride) (PVDF)-based electrolytes with adequate mechanical strength, good thermal stability and high ionic conductivity have recently become particularly attractive[11–14]. Little amount of N,N-dimethylformamide (DMF) solvent have been identified to exist in the electrolyte, due to the strong interactions between Li salt and solvent. The polar DMF solvent combined with high dielectric constant ($\varepsilon_r$, 8–12) of the PVDF polymer facilitates dissociation of Li salt to form $[Li(DMF)_x]^+$ solvation structures, which can be transported by polymer chains based on DMF-PVDF interactions[15–19]. Importantly, the PVDF-based

[1]State Key Laboratory of Chemical Engineering, Institute of Pharmaceutical Engineering, College of Chemical and Biological Engineering, Zhejiang University, 310027 Hangzhou, Zhejiang, China. [2]ZJU-Hangzhou Global Scientific and Technological Innovation Center, Zhejiang University, 311215 Hangzhou, China. [3]College of Chemical and Biological Engineering, Zhejiang University, 310058 Hangzhou, Zhejiang, China. [4]School of Future Technology, Department of Chemical Physics, and Anhui Center for Applied Mathematics, University of Science and Technology of China, 230026 Hefei, China. [5]Innovation Research Institute of Technology Center, Zhejiang Xinan Chemical Industrial Group Co. ltd, 311600 Hangzhou, Zhejiang, China. [6]Department of Chemical Engineering, University of Washington, Seattle, WA 98195, USA. ✉e-mail: yingyinglu@zju.edu.cn

electrolytes deliver high ionic conductivity of $10^{-4}$ S cm$^{-1}$, enabling solid-state LMBs to operate at room temperature.

Nevertheless, PVDF-based electrolytes face many serious challenges. The electrolyte possesses a porous structure due to phase separation between the polymer and solvent[20,21], which results in heterogeneous ion flux across the electrolyte when paired with the Li metal anode, leading to rapid Li dendrite growth and short circuit of the batteries[21,22]. Although DMF solvent plays a key role in ion transport, it also brings about several issues. Its side reactions with Li metal and poor antioxidation ability lead to continuous decomposition at the interfaces and narrow electrochemical stability window of the electrolyte[23–25]. In addition, the ionic conductivity of PVDF-based electrolytes is still far from that of LEs for practical applications. Addressing these issues is highly expected to realize high-performance room-temperature solid-state LMBs with PVDF-based electrolytes.

Great efforts focusing on enhancing interfacial compatibility and ionic conductivity have been made. Employing electrolyte additives[26–28], adjusting the types of Li salt and solvent[29–31], regulating the solvent content[17], and anchoring the solvent with fillers[21] have been reported to effectively suppress side reactions. Xu et al. reported that PVDF-based electrolytes possess a local high concentrated (LHC) structure, in which a high concentrated solution formed by lithium bis(trifluoromethanesulfonyl) (LiTFSI) and dimethyl sulfoxide (DMSO) solvent is embedded inside polymer spherulites, and the LHC structure of the PVDF electrolyte can mitigate interfacial side reactions due to the interactions between solvent and PVDF polymer[32]. However, solvent decomposition at high current density (above 0.5 mA cm$^{-2}$), high rate (above 1C) and high potential (above 4.3 V) could be accelerated, which results in limited cycle life and low capacity retention of the batteries. In terms of increasing ionic conductivity of the electrolyte, many studies introduced active and negative fillers (hydroxide[33], Li$_{0.33}$La$_{0.56}$TiO$_{3-x}$[34]) or modified polymer molecule structures[35] to decrease the crystallinity degree of PVDF. However, the PVDF polymer itself might not be the critical factor due to its rigid property. As a result, the ionic conductivity is unsatisfactory. What's worse, obtaining dense PVDF-based electrolytes towards their practical application remains a great challenge. Therefore, proposing an innovative strategy for developing dense PVDF-based electrolytes with superior ion transport capability and stable interfaces against Li metal anodes and high-voltage cathodes is of both fundamental and technological importance.

Herein, we design a PVDF-based composite electrolyte by ingeniously adding two-dimensional (2D) transition metal dichalcogenide MoSe$_2$ sheets (MSs). The broken inversion symmetry of MoSe$_2$ can strongly interact with the dipole moment of the PVDF monomer units, in which the Mo atom with a positive charge interacts with -CF$_2$- and the Se atom with a negative charge interacts with -CH$_2$- (Fig. 1a). This interaction not only promotes the all-trans ($\beta$-phase) transformation of PVDF to enhance the $\beta$-phase percentage, but also forms a dense composite electrolyte after adding 15 wt.% MSs (denoted PVMS-15). Then, the $\beta$-phase-rich electrolyte delivers a higher dielectric constant, with $\varepsilon_r$ increasing from 9.6 to 21.1, which optimizes the solvation structures to form solvent-separated ion pair (SSIP) (Fig. 1b), achieving a higher ionic conductivity ($6.5 \times 10^{-4}$ S cm$^{-1}$) with a lower activation energy (0.07 eV) of the PVMS-15 electrolyte. Furthermore, the in situ reactions between MoSe$_2$ and Li metal generate a Li-conducting Li$_2$Se component in solid electrolyte interphase (SEI), which could suppress the DMF decomposition, improve the Coulombic efficiency (CE) and enhance the interfacial ion transport kinetics (Fig. 1c). With the above merits, Li||Li symmetric cells deliver robust cycling of 480 h at 1 mA cm$^{-2}$, and Li||NCM811 full cells show practical performance at high current density (3C), high loading (2.6 mAh cm$^{-2}$) and in pouch cell. This work provides a practical electrolyte engineering strategy for the development of stable solid-state LMBs with ultralong lifespans.

## Results
### Characteristics of PVDF and PVMS-based electrolytes

The MSs were synthesized via an in situ selenization route[36]. The X-ray diffraction (XRD) patterns in Supplementary Fig. 1 show that the crystal with a hexagonal structure corresponds to single-phase 2H-MoSe$_2$ (PDF #29-0914). Free-standing and flexible PVDF and PVMS-15 composite electrolytes were prepared by a solution-casting method using DMF solvent and LiFSI (Supplementary Fig. 2). After being blended with the PVDF electrolyte, the structure of the MSs can be well maintained (Fig. 1d). The surface scanning electron microscopy (SEM) image shows that the MSs are uniformly dispersed to form a dense structure, in which the PVDF spherulites present strong adhesion with MSs due to the interactions (Fig. 1e and Supplementary Fig. 3). In contrast, the PVDF electrolyte exhibits a porous structure due to the phase separation between the polymer and solvent[20,31] (Fig. 1f and Supplementary Fig. 4). As a result, the tensile strength of the PVMS-15 electrolyte increases from 0.6 to 1.28 MPa, and the Young's modulus of the PVMS-15 electrolyte (1539 MPa) is much higher than that of the PVDF electrolyte (213 MPa), as shown in Supplementary Fig. 5. The cross-sectional images show that the thicknesses of PVDF and PVMS-15 electrolytes are 100 μm and 80 μm (Supplementary Fig. 6), respectively. These results suggest that a dense structure and enhanced mechanical properties of the PVMS-15 electrolyte can be achieved by adding the MSs filler.

Considering that the DMF solvent plays a key role in ion transport and SEI formation[18], we explored the influence of dense structure on solvent distribution in PVMS-15 electrolyte. Thermogravimetric analysis (TGA) measurements show that the contents of the DMF solvent for PVMS-15 and PVDF electrolytes are -12.46 wt.% and 13.95 wt.% (Supplementary Fig. 7), respectively, which are consistent with the solid-state nuclear magnetic resonance (ss-NMR) test results (Supplementary Fig. 8), indicating that the addition of MSs does not affect the solvent content. We conducted atomic force microscopy-nano-infrared spectroscopy (AFM-nano-IR) to detect the C=O group of DMF in the PVMS-15 and PVDF electrolytes. The intensity of the absorption peak remains low and consistent on the surface of the PVMS-15 electrolyte (Fig. 1g, h), which is beneficial for forming a dense SEI and suppressing DMF decomposition. In contrast, the DMF solvent aggregated around PVDF spherulites (Supplementary Fig. 9 and Fig. 1i), which always induces uneven Li deposition and rapid Li dendrite growth, as evidenced by previous reports[21,22] and Supplementary Note 1. Therefore, the adsorption between MSs and PVDF not only generates a dense electrolyte, but also results in a uniform solvent distribution.

To determine this interaction between PVDF and MSs, density functional theory (DFT) calculations were first conducted. PVDF has repeated units of -CH$_2$-CF$_2$- monomers in its polymer chain. Each monomer unit of PVDF has a strong dipole moment due to the electronegativity of F atoms compared to H and C atoms[37,38]. Specifically, $\alpha$-phase PVDF consists of alternate trans and gauche conformations (TGTG), which are nonpolar due to the antiparallel arrangement of dipoles. In comparison, $\beta$-phase PVDF possesses the all-trans conformation (TTTT), which presents the highest polarity because of the parallel arrangement of dipoles for each monomer unit. Interestingly, we found that when an $\alpha$-phase PVDF chain was initially adsorbed on the surface of the MoSe$_2$ crystal, its structure could be fully transformed into $\beta$-phase PVDF after geometry optimization, demonstrating strong interactions (Supplementary Figs. 12–17). In addition, the Mo atom with a positive charge interacts with -CF$_2$-, while the Se atom with a negative charge interacts with -CH$_2$-, due to the asymmetry property of MSs and the strong dipole moment of PVDF. To confirm the enhancement of $\beta$-phase PVDF in the developed PVMS composite electrolytes, Fourier transform infrared (FTIR) spectroscopy was carried out. As shown in Fig. 2a, the peaks corresponding to the $\beta$-phase PVDF are marked at 840 and 1388 cm$^{-1}$, and the peaks corresponding

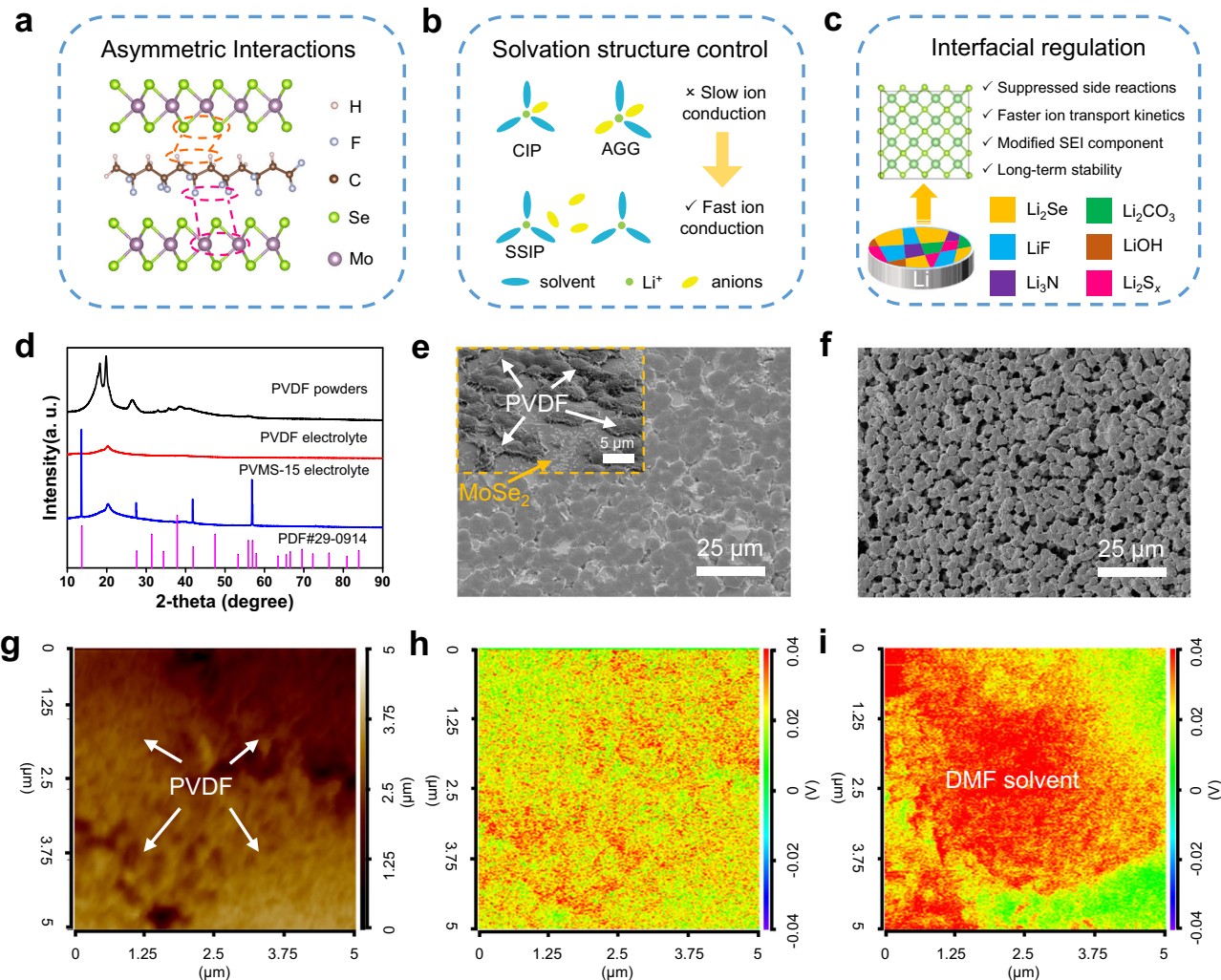

**Fig. 1 | Design principles and structural characteristics of the electrolytes.**
**a** Asymmetric interactions. **b** Solvation structure control. **c** Interfacial regulation. **d** XRD patterns of PVDF powders, PVDF and the PVMS-15 electrolytes. **e** Surface SEM image of the PVMS-15 electrolyte. The inset is surface of PVMS-15 electrolyte obtained from FIB-SEM. **f** Surface SEM image of the PVDF electrolyte. **g** AFM and nano-IR overlap of the C = O vibration of DMF solvent in the PVMS-15 (**h**) and PVDF (**i**) electrolytes. The red color area means the existence of DMF solvent while the green color area means the absence of DMF solvent.

to 762 and 1360 cm⁻¹ are related to the α-phase PVDF. The average content of the β-phase is calculated to be 39% using the Lambert-beer law for the PVDF electrolyte[38] (Eq. 3 in "Methods", details in Supplementary Fig. 18 and Supplementary Table 1). In contrast, with increasing amounts of MSs, the ratio of β-phase PVDF increases to 52%, 64%, and 77% for the PVMS-10, PVMS-15, and PVMS-20 electrolytes, respectively. However, aggregation of MSs was observed for the PVMS-20 electrolyte (Supplementary Fig. 19), which resulted in an uneven morphology and poor interfacial contact, so the optimal content of MoSe₂ was regulated to be 15 wt.% to obtain a dense and flat PVMS-15 electrolyte. The above findings prove that the interactions are attributed to asymmetric adsorption between MSs and the dipole moment of PVDF monomer units.

### Physicochemical and electrochemical properties of the electrolytes

To evaluate the significance of the dense structure and associated changes in the electrolyte environment, we tested physicochemical and electrochemical properties. As shown in Fig. 2b, c and Supplementary Fig. 20, the dielectric constant ($\varepsilon_r$) increases from 9.6 to 21.1 due to the higher content of β-phase PVDF, and there is a positive correlation between the ratio of β-phase PVDF and $\varepsilon_r$ (Fig. 2d). Recently, $\varepsilon_r$ has been widely considered to be a critical parameter to

promote Li salt dissociation and ion transport in SSEs[15,16,39], which motivated us to reveal the inherent solvation structures via FTIR and Raman spectroscopy. All the DMF molecules are bound to Li⁺ due to the absence of a band at 658 cm⁻¹ for free DMF in the FTIR spectra[17] (Fig. 2a). The S-N-S band of FSI⁻ in the Raman spectra consists of four modes: SSIP (719 cm⁻¹), CIP (730 cm⁻¹), AGG-1 (742 cm⁻¹) and AGG-2 (750 cm⁻¹)[40-42], and undissociated LiFSI is not present in the electrolyte[32]. In detail, the PVDF electrolyte contains 15.21% SSIP, 40.53% CIP, 32.69% AGG-1 and 11.57% AGG-2 (Fig. 2e and Supplementary Fig. 21). Noticeably, in the case of the PVMS-15 electrolyte, the proportions of AGG-1 and AGG-2 decrease to 17.74% and 10.28%, respectively, while the amount of SSIP dramatically increase, suggesting that solvation structures can be changed by the addition of MSs. We also conducted ⁷Li and ¹⁹F nuclear ss-NMR measurements to verify this phenomenon. Compared with the PVDF electrolyte, both the ⁷Li and ¹⁹F peaks shift downfield in the PVMS-15 electrolyte, indicating weakened FSI⁻-Li⁺ interactions and strengthened Li⁺-DMF coordination[43,44], which is consistent with the Raman spectra results (Fig. 2f, g). Therefore, the β-phase-rich PVMS-15 electrolyte could change the inherent solvation structures to form abundant SSIPs, contributing to faster ion transport in the electrolyte.

As expected, the PVMS-15 electrolyte exhibits much higher ionic conductivity ($6.4 \times 10^{-4}$ S cm⁻¹) than the PVDF electrolyte

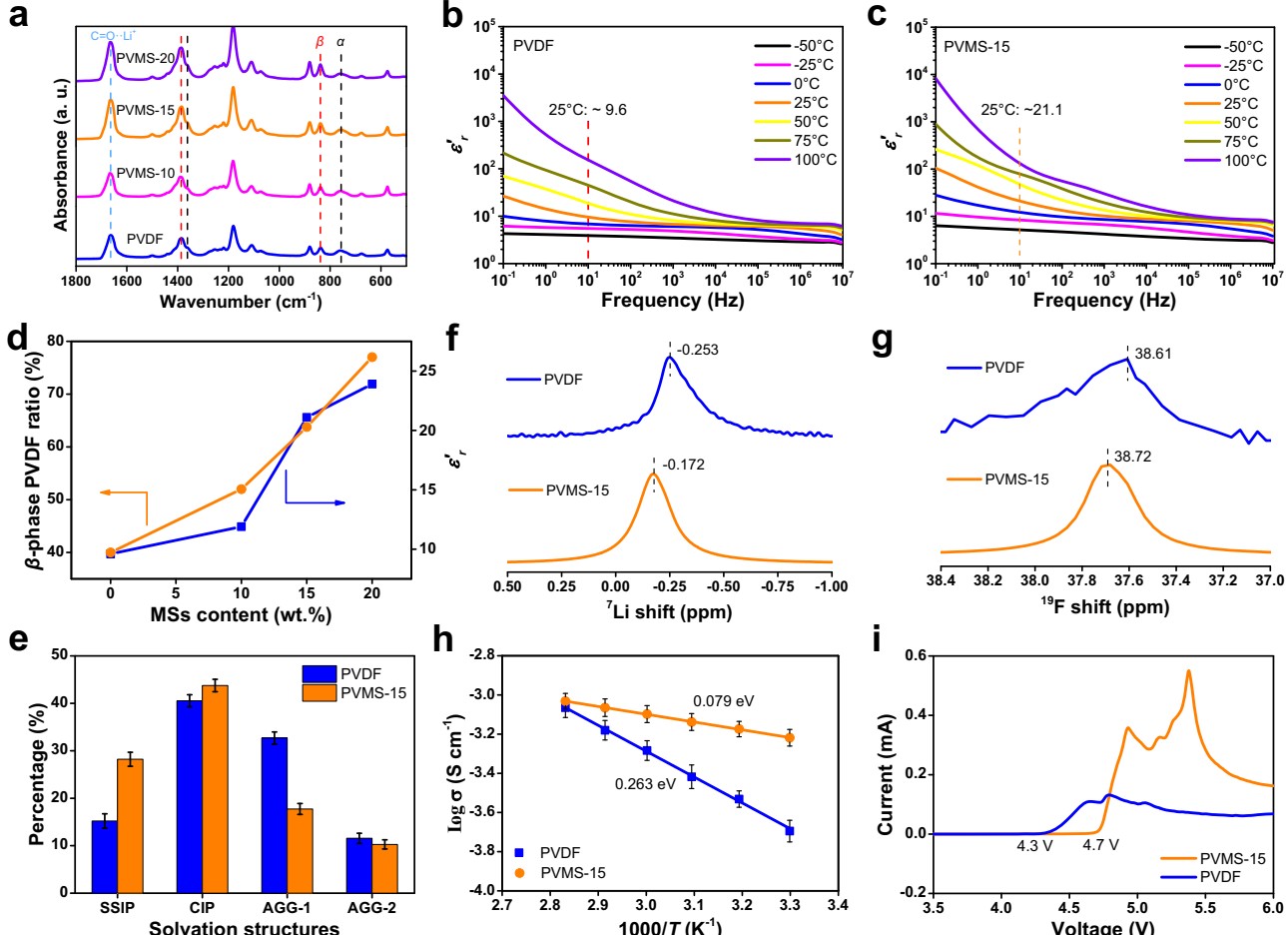

**Fig. 2 | Solvation structure analysis and electrochemical properties of the electrolytes. a** FTIR spectra of the electrolytes. Real part ($\varepsilon_r$') of the relative permittivity as a function of frequency at different temperatures for the PVDF (**b**) and PVMS-15 (**c**) electrolytes. **d** Relationship between the MSs content and $\beta$-PVDF ratio and $\varepsilon_r$'. **e** Raman spectra results of the PVDF and PVMS-15 electrolytes. Values are means, and error bars were calculated by taking the standard errors from the measurements with three identical samples. **f** $^7$Li NMR spectra. **g** $^{19}$F NMR spectra. **h** Arrhenius plots of the ionic conductivities of the PVMS-15 and PVDF electrolytes. Values are means, and error bars were calculated by taking the standard errors from the measurements with three identical samples. **i** LSV curves of the PVMS-15 and PVDF electrolytes.

($2.1 \times 10^{-4}$ S cm$^{-1}$) (Supplementary Fig. 22). The Arrhenius plots show that the activation energy ($E_a$) for ion transport decreases from 0.26 to 0.07 eV (Fig. 2h). It is also worth noting that the MS as a semiconductor material shows a negligible influence on the electronic conductivities of the electrolytes to meet the requirements of practical batteries (Supplementary Fig. 23 and Supplementary Tables 2 and 3). The linear sweep voltammetry (LSV) curves show that the electrochemical stability window can be extended to 4.7 V (Fig. 2i), indicating that DMF decomposition could be suppressed in the dense PVMS-15 electrolyte. As discussed, phase regulation can not only obtain dense electrolyte, but also change the inherent solvation behavior of the PVMS-15 electrolyte to deliver enhanced ion transport capability and electrochemical properties.

## Li metal compatibility characterizations

When transition metal dichalcogenides (TMDs) MSe$_x$ (M = W, Nb, Mo, etc.) are applied in LMBs, they can easily form Li$_2$Se in the SEI derived from the irreversible redox reaction of Li metal and MSe$_x$[45–49]. Thus, we systematically characterized the morphology and constituents of the SEI at the atomic level by cryogenic scanning transmission electron microscopy (cryo-STEM). The sample was prepared by plating Li on a Cu grid with a current density of 0.1 mA cm$^{-2}$ for 6 h in a Cu||Li cell. From the cryo-TEM analyses, we observed a smooth and dense SEI (13 nm) formed by the PVMS-15 electrolyte, while the SEI formed by the

PVDF electrolyte is much rougher and thicker (Supplementary Fig. 24). After matching the fast Fourier transform (FFT) patterns of the corresponding species with the known lattice planes, we confirmed that the SEI exhibits a classical mosaic structure consisting of an amorphous matrix and embedded Li$_2$O, Li$_2$CO$_3$, LiOH, and Li$_2$Se inorganic crystals[50,51] (Fig. 3a, b and Supplementary Fig. 25). More notably, Li$_2$Se nanoparticles with a lattice corresponding to the (111) and (200) planes could be clearly detected[52] (Fig. 3c, d). In contrast, much more LiOH, Li$_2$O, and Li$_2$CO$_3$ could be observed in the SEI formed by the PVDF electrolyte, due to the severe decomposition of DMF (Supplementary Fig. 26). We also conducted the cyclic voltammetry (CV) tests of Li||Cu cells to demonstrate the in situ formation of Li$_2$Se. As shown in Supplementary Fig. 27, the cathodic peak at 0.8–1.5 V during the first cycle corresponds to the intercalation of Li$^+$ into the MSs to form Li$_2$MoSe$_2$, and the peak at 0.15 V can be assigned to the reduction of Li$_x$MoSe$_2$ to form metal Mo and Li$_2$Se[53]. It should be noted that the formation potential of Li$_2$Se is higher than DMF reduction potential on Li metal[24,54], which could also suppress side reactions and assist to enhance the electrochemical stability of the PVMS-15 electrolyte. As a result, the current response with the PVMS-15 electrolyte increases during Li plating/stripping on the Cu substrate, corresponding to rapid Li$^+$ transport through the Li$_2$Se-containing SEI. The good peak reversibility indicates a highly stable property. In sharp contrast, the CV curves obtained using the PVDF electrolyte deliver a much smaller

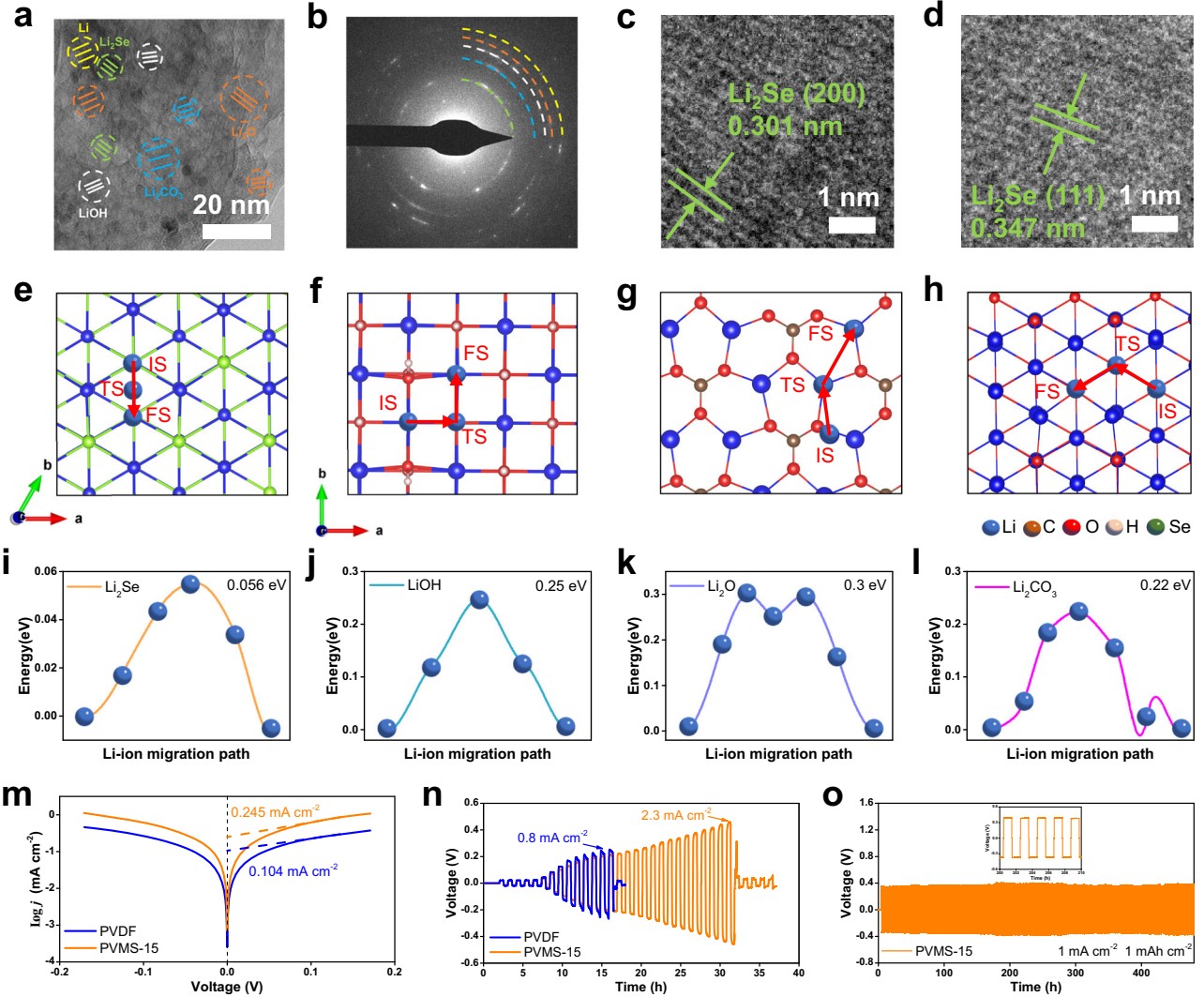

**Fig. 3 | Interfacial analysis and characterizations. a** Enlarged TEM image of the SEI formed by the PVMS-15 electrolyte. **b** FFT pattern. **c** HRTEM image of $Li_2Se$ (200). **d** HRTEM image of $Li_2Se$ (111). Ion diffusion pathways and barriers of (**e**, **i**) $Li_2Se$, (**f**, **j**) LiOH, (**g**, **k**) $Li_2O$ and (**h**, **l**) $Li_2CO_3$. **m** ECD test. **n** CCD test. **o** Galvanostatic cycling curves of Li‖Li cells with the PVMS-15 electrolyte at a current density of $1\,mA\,cm^{-2}$.

current response and poor peak reversibility, demonstrating continuous side reactions and an unstable SEI.

To further evaluate the influence of $Li_2Se$ on the ion transport kinetics in the SEI, we performed calculations to reveal the ion transport pathways and barriers of several components, including LiOH, $Li_2CO_3$, and $Li_2O$. The lattice parameters are provided in Supplementary Table 3, and the most stable interface of each crystal was selected for calculations based on the surface energies (Supplementary Table 4). We investigated the adsorption behavior of the Li adatoms on the crystal surfaces to determine the initial and final states of the diffusion path. The adsorption energy was calculated to account for the adhesion strength of adsorbates. The top sites are proven to be the energetically favorable adsorption sites for $Li_2Se$ and $Li_2O$, as shown in Fig. 3e, g, respectively. The adsorbate Li atoms are directly above the Se and O sites due to their high electronegativity. For LiOH, the bridge sites between two O-top atoms were found to be the preferred adsorption position. For $Li_2CO_3$, this position is the bridge site of one O-top atom and one O-hcp atom, as shown in Fig. 3e, i. The transition state of $Li_2Se$ is the adsorption of Li atoms at the bridge sites of Li-hcp and Se-top. As a result, the Li atom maintains a migration energy of 0.056 eV to diffuse on $Li_2Se$ (111). In contrast, LiOH (001), $Li_2O$ (111) and $Li_2CO_3$ (002) have to overcome large energy barriers of 0.25 (Fig. 3f, j),

0.3 (Fig. 3g, k) and 0.22 eV (Fig. 3h, l), respectively. Given this, we can conclude that $Li_2Se$ works as a fast conductor in the SEI to enhance the ion transport kinetics, contributing to robust cycling of the batteries.

We then cycled Li‖Li cells to assess the properties of the $Li_2Se$-containing SEI. As shown in Fig. 3n and Supplementary Fig. 28, the Li|PVMS-15|Li cell enables critical current densities (CCDs) of $2.3\,mA\,cm^{-2}$ and $8.3\,mA\,cm^{-2}$ using the time control and capacity control methods[55], respectively, which are much higher than those of the Li|PVDF|Li cell ($0.8\,mA\,cm^{-2}$ and $1.6\,mA\,cm^{-2}$), indicating that Li dendrite growth can be effectively suppressed. The higher exchange current density (ECD) of $0.245\,mA\,cm^{-2}$ indicates enhanced ion transport kinetics of the SEI (Fig. 3m). The Li|PVMS-15|Li cell can stably cycle for 1900 h at $0.1\,mA\,cm^{-2}$ with a smaller polarization voltage, while the Li|PVDF|Li cell displays short circuit after 290 h (Supplementary Fig. 29). In addition, the Li|PVMS-15|Li cell is capable of cycling at $1\,mA\,cm^{-2}$ for lifespan of 480 h (Fig. 3o). The critical deposition capacity (CDC) achieves as high as $2.6\,mAh\,cm^{-2}$ (Supplementary Fig. 30), indicating the compatibility with high-loading cathode in the Li‖NCM811 full cells. To evaluate the stability of the SEI and its influence on Li deposition morphology, we assembled the Li‖Cu cells. As shown in Supplementary Fig. 31, the Li|PVMS-15|Cu cell delivers the cycle life of 55 cycles with an average CE as high as 97.8%, while the Li|PVDF|Cu cell presents

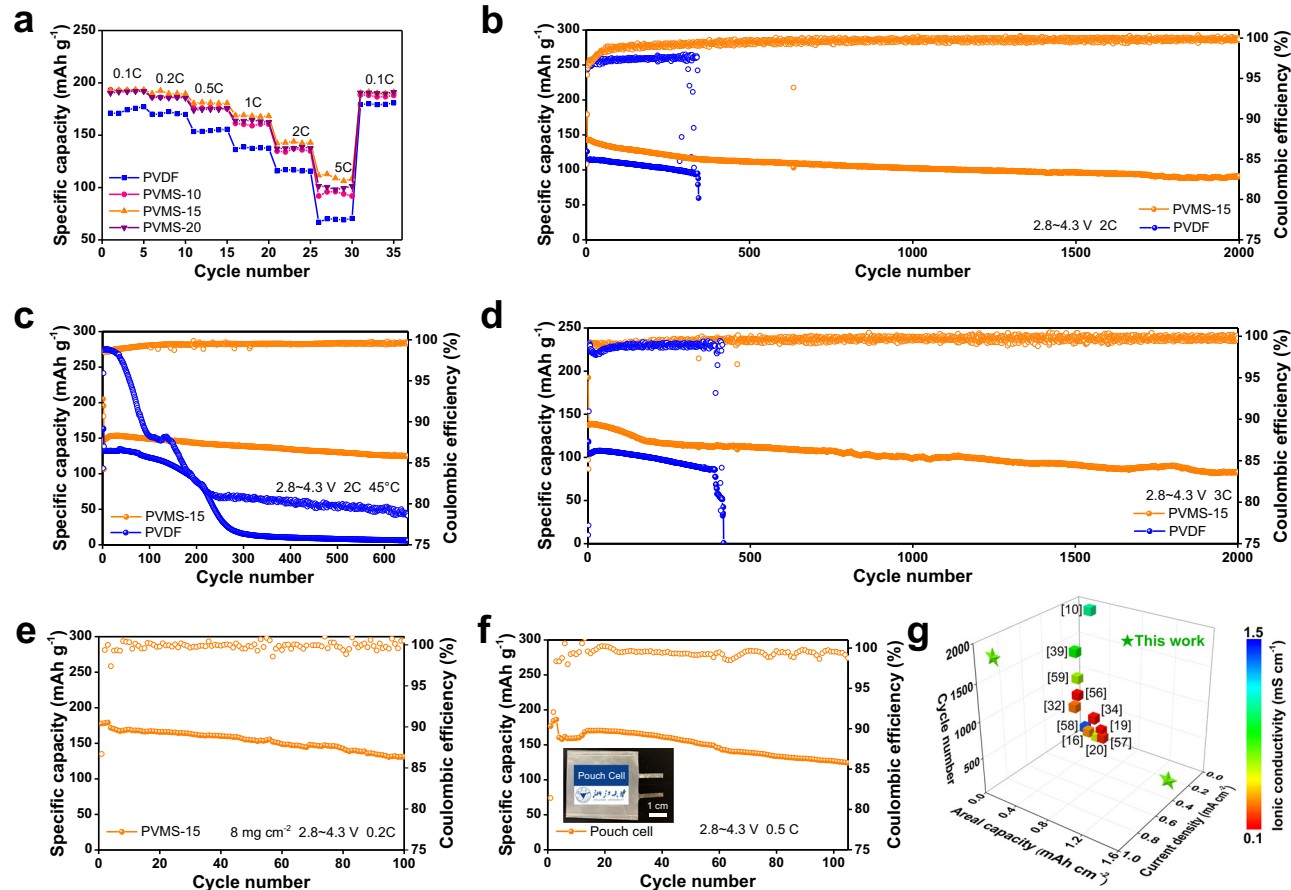

**Fig. 4 | Electrochemical performance of Li∥NCM811 full cells. a** Capacities at various current density. Long-term cycling stability at 2.8–4.3 V, 2C, 2 mg cm⁻² and 25 °C (**b**), 2.8–4.3 V, 2C, 2 mg cm⁻² and 45 °C (**c**), 2.8–4.3 V, 3C, 2 mg cm⁻² and 25 °C (**d**), 2.8–4.3 V, 0.1C, 8 mg cm⁻² and 25 °C (**e**) and the pouch cell (**f**). 1C is defined as 180 mA g⁻¹. **g** Comparison of the electrochemical performance of the device in this work with that of other recently reported solid-state batteries using PVDF-based and other polymer-based electrolytes.

a short circuit after only 12 cycles with a low average CE of 86.4%. After plating 1 mAh cm⁻² Li on the Cu foil, dense and connected Li particles can be observed with the PVMS-15 electrolyte (Supplementary Fig. 32), while small and uneven Li particles are induced by the PVDF electrolyte (Supplementary Fig. 10). These results prove that the formation of Li₂Se in the SEI can significantly enhance the Li metal compatibility.

## Electrochemical performances of Li∥NCM811 full cells

To further prove how phase regulation and enhanced interfacial ion transport kinetics influence electrochemical performance, we tested the Li∥NCM811 full cells under comprehensive operation conditions. As shown in Supplementary Fig. 33, CV curves show that the Li|PVMS-15|NCM811 cell displays better redox reaction reversibility and smaller lithiation/delithiation polarization, resulting in a smaller charge transfer resistance ($R_{ct}$) (Supplementary Fig. 34). At the coin cell level (450 μm Li foil, 2 mg cm⁻² NCM811 loading), discharge capacities of 193.4, 169.2, 144, and 112.9 mAh g⁻¹ can be observed at rates of 0.1, 1, 2, and 5C, respectively, which are higher than those obtained using the PVMS-10, PVMS-20 and PVDF electrolytes (Fig. 4a and Supplementary Fig. 35). After the rate tests, the $R_{ct}$ obtained using the PVMS-15 electrolyte increases from 35 to 50 Ω, while that using PVDF electrolyte dramatically increases from 60 to 180 Ω (Supplementary Fig. 36). In terms of long cycling stability, the Li|PVMS-15|NCM811 cell shows an ultralong lifespan of 2000 cycles at 2C with a higher average CE of 99.7%, while the Li|PVDF|NCM811 cell presents a short circuit with a lower CE of 97.5% after only 341 cycles (Fig. 4b). At a higher rate of 3C, the Li|PVMS-15|NCM811 cell can also stably cycle 2000 times with a

capacity retention of 60.3% (Fig. 4d). Remarkably, the cycling stability of the Li|PVMS-15|NCM811 cells is better than that of the commercial liquid cells under the same loadings (Supplementary Fig. 37). Such improvements demonstrate the good ion transport capability and electrochemical stability of the PVMS-15 electrolyte.

As a further step, the performance under many harsh conditions, including high cutoff voltage, wide temperature range, and pouch cell, was evaluated. As shown in Fig. 4c, at an elevated temperature of 45 °C, the Li|PVMS-15|NCM811 cell shows a high capacity retention of 85.7% after 660 cycles, while the Li|PVDF|NCM811 cell suffers from rapid capacity fading after 100 cycles with a much lower CE of 81.3%, which could be attributed to severe electrolyte decomposition. Under a higher cutoff voltage of 4.5 V, the Li|PVMS-15|NCM811 can still normally cycle 600 times with a capacity retention of 77.8%, while overcharge can be observed in the initial cycle for the Li|PVDF|NCM811 cell (Supplementary Fig. 38). Moreover, under practical cathode active material loadings of 1.44 mAh cm⁻² and 2.6 mAh cm⁻², the Li|PVMS-15|NCM811 cell presents lives of 100 and 25 cycles (Fig. 4e and Supplementary Fig. 39), respectively. Contributing to high ionic conductivity and low activation energy for ion transport, the Li|PVMS-15|NCM811 cell achieves cycling at −20 °C for 150 cycles (Supplementary Fig. 40). To the best of our knowledge, our developed cells outperform most of the reported SSBs using PVDF-based and other polymer-based SSEs[10,16,19,20,32,34,39,56–59] (Fig. 4g and Supplementary Table 5). These results also reinforce our conclusion that densifying the electrolyte and regulating the ion transport are critical in boosting the electrochemical performance of PVDF-based electrolytes.

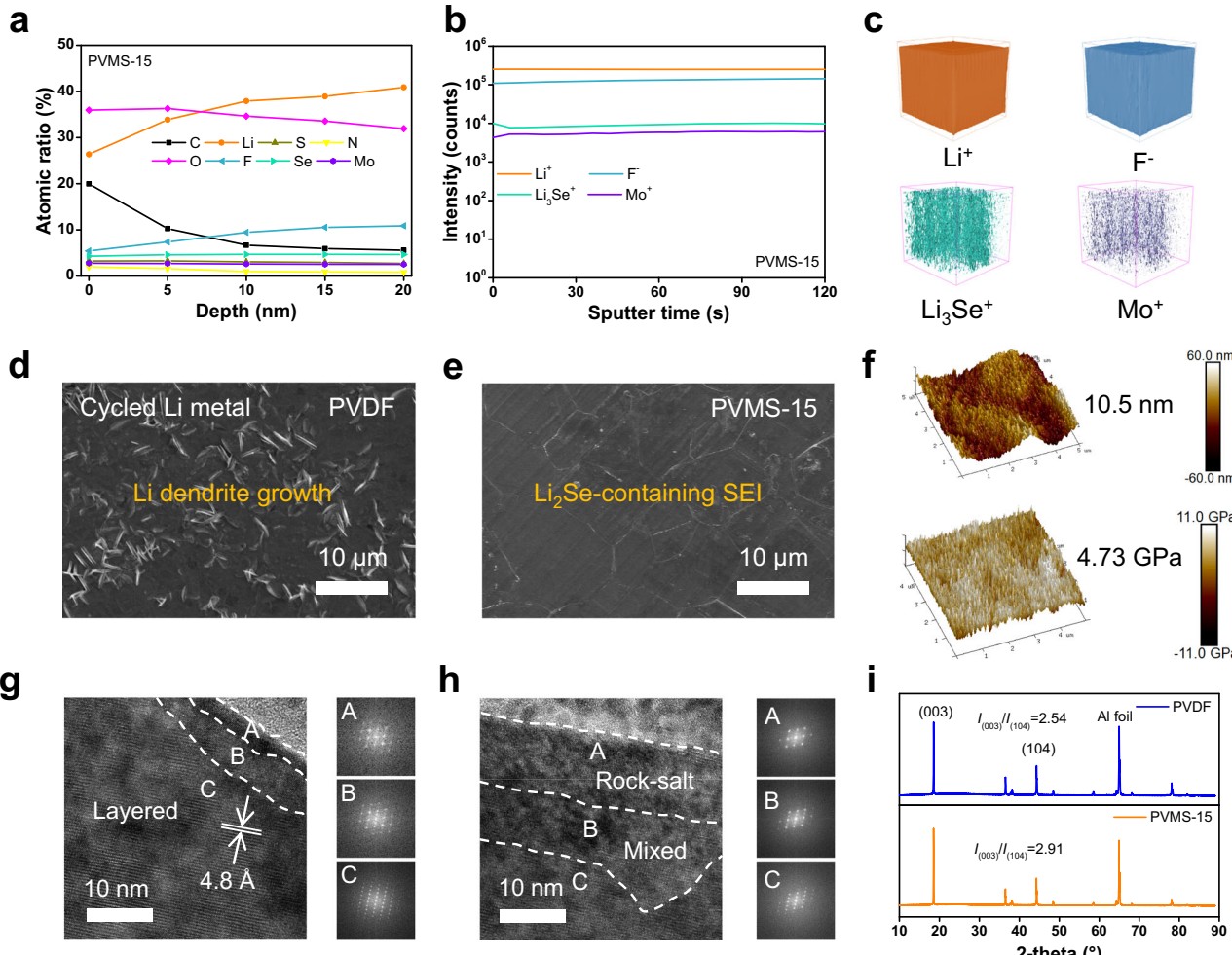

**Fig. 5 | Interfacial and structural analysis of the cycled electrode. a** Atomic ratios of the SEI at different sputter time with the PVMS-15 electrolyte. **b** TOF-SIMS depth profiles of the secondary ion fragments and 3D view (**c**) in the formed SEI by PVMS-15 electrolyte. Surface SEM images of the cycled Li metal using the PVDF (**d**) and PVMS-15 (**e**) electrolytes. **f** 3D AFM image of the roughness and Young's modulus tests of cycled Li metal when using the PVMS-15 electrolyte. Nanostructure of the cycled NCM811 when using the PVDF (**g**) and PVMS-15 (**h**) electrolytes. **i** XRD spectra of the cycled NCM811 cathodes.

Furthermore, we explored the behavior of single-layer pouch cells (cell-1: 2 mg cm$^{-2}$ loading, 4 cm × 4.5 cm, 20 μm Li foil; cell-2: 8 mg cm$^{-2}$, 3 cm × 3.5 cm, 50 μm Li, negative to positive areal capacity (N/P) ratio = 6.25, 15 mAh capacity). The pouch cell-1 exhibits a discharge charge capacity of 170 mAh g$^{-1}$ and a capacity retention of 78.5% after 110 cycles (Fig. 4f and Supplementary Fig. 41), and the pouch cell-2 can stably cycle for 20 times at 0.1C (Supplementary Fig. 42), demonstrating the application potential of the PVMS-15 electrolyte. The pouch cell-1 can still work after folding, puncturing, and cutting tests (Supplementary Fig. 43).

**Interfacial and structural analysis of the cycled electrode**
As the last piece of the puzzle, understanding the impact of phase regulation and associated interfacial regulation on the electrodes is important. All the tests were conducted after cycling Li||NCM811 cells 20 times at 25 °C. The X-ray photoelectron spectroscopy (XPS) results show that the SEI mainly consists of C-containing organic species and inorganic species, including $Li_2CO_3$, $Li_2O$, LiF, and sulfur compounds (Supplementary Fig. 44), which are attributed to the decomposition of DMF solvent and FSI$^{-29}$, respectively. $Li_2Se$ and Mo can be clearly detected, demonstrating that reactions between MSs and Li metal can occur. As shown in Fig. 5a and Supplementary Fig. 45, the C and O atomic contents are decreased with the PVMS-15 electrolyte, demonstrating that DMF decomposition is effectively

mitigated. The Se and Mo contents at different sputtering times remain constant at approximately 4.68% and 2.53%, respectively, indicating a uniform distribution of $Li_2Se$ and Mo. We also performed time-of-flight secondary ion mass spectrometry (TOF-SIMS) measurements to evaluate the composition in a 200 μm × 200 μm × 50 nm area. As shown in Fig. 5b and Supplementary Fig. 46, many organic species (CN$^-$, C$^-$, and CNO$^-$) and inorganic species ($Li_3OH^+$, $Li_3CO_3^+$, $Li_3O^+$, $Li_3Se^+$, Mo$^+$, and F$^-$) can be obtained, which is consistent with the XPS results. In detail, the intensity profiles and corresponding three-dimensional (3D) views indicate that inorganic species are uniform and dominant in the SEI. As a result, the cycled Li metal when using the PVMS-15 electrolyte presents a flat and uniform morphology (Fig. 5e), as evidenced by the uniform distributions in the F, N, and Se element mappings (Supplementary Fig. 47), while many Li dendrites and disconnected SEI can be observed when using the PVDF electrolyte (Fig. 5d). Due to the formation of $Li_2Se$ and suppressed decomposition of DMF, the SEI formed by the PVMS-15 electrolyte delivers a higher Young's modulus (4.73 GPa) and lower surface roughness (10.5 nm) than that formed by the PVDF electrolyte (1.54 GPa and 16.3 nm), as shown in Fig. 5f and Supplementary Fig. 48. Furthermore, it is essential to note that despite the metallic Mo and Mo-containing species are formed, they show negligible influence on the electronic conductivity of the SEI (Supplementary Fig. 49 and Note 2). Therefore, the dense structure of the PVMS-15

electrolyte accompanied by the interfacial regulation contributes to robust cycling and Li metal compatibility.

In terms of the cathode electrolyte interphase (CEI) for cycled NCM811, the lower C and O atomic contents demonstrate suppressed electrolyte oxidation during cycling (Supplementary Figs. 50 and 51). The TEM images show that a uniform CEI with a thickness of 4 nm is formed when using the PVMS-15 electrolyte, which is thinner than that with a thickness ~7.5 nm formed by using the PVDF electrolyte (Supplementary Fig. 52). The nanostructures of cycled NCM811 particles show rock-salt phase, mixed phase and layered structure (Fig. 5g, h). The high-resolution transmission electron microscopy (HRTEM) image shows a thick disordered rock-salt phase layer (10 nm) on the cycled NCM811 when using the PVDF electrolyte, while the irreversible phase transformation can be mitigated with the PVMS-15 electrolyte due to the enhanced cathode/electrolyte compatibility, which can also be proven by the higher $I_{(003)}/I_{(104)}$ from the XRD results (Fig. 5i). In general, the oxidation of the PVDF electrolyte leads to thick CEI and structural degradation of NCM811, resulting in capacity fade and large polarization. As discussed above, the dense structure combined with the enhanced ion transport capability and superior interfacial stability of PVMS-15 electrolyte boost the electrochemical performance toward practical applications.

## Discussion

In summary, we develop a PVDF-based composite electrolyte with a dense structure, enhanced ion transport capability and interfacial stability, which achieves robust cycling under practical conditions including high rate (3C), high loading (2.6 mAh cm$^{-2}$) and in pouch cells. The interactions between MSs and the dipole moment of PVDF monomer units can disrupt the symmetry of PVDF to promote its $\beta$-phase transformation, which can further form a high dielectric environment in the electrolyte to tailor the solvation structures, endowing high ionic conductivity and low activation energy. In addition, the in situ reactions between MoSe$_2$ and Li metal generate the fast conductor Li$_2$Se in the SEI, which improves the CE and enhances the interfacial kinetics. This work not only addresses several key issues of PVDF-based electrolytes through an ingenious design, but also provides an encouraging strategy that contributes to low-cost and large-scale production toward their practical applications.

## Methods

### Preparation of the MoSe$_2$ nanosheets

(NH$_4$)$_6$Mo$_7$O$_{24}$·4H$_2$O, Se, and Mg metal powders with weight ratio of 0.3:1:1 were blended within an agate mortar, which were then transferred into a stainless-steel autoclave and heated in a pot furnace with raising the temperature to 800 °C for 4 h followed by natural cooling. The obtained gray powder was washed with dilute hydrochloric acid (HCl) in a 10:7 ratio to eliminate Mg and MgO residues. Subsequently, multiple washes with distilled water were performed to remove remaining traces of HCl. Finally, the powder was dried at 110 °C in a hot air oven.

### Preparation of the PVDF and PVMS electrolytes

PVDF powders were dried at 60 °C for 12 h before use. The PVDF and PVMS electrolytes were prepared by a solution-casting method in a culture dish (diameter: 100 mm) with a PVDF/LiFSI weight ratio of 3/2 using the DMF solvent. The weight percentage of the MSs in the mass of PVDF was 10%, 15%, and 20%. The solid-state PVDF and PVMS electrolytes were obtained by drying for 24 h at 55 °C and stored in a glovebox.

### Materials characterization

X-ray diffraction (XRD) tests were conducted in Bruker D8 Advance with Cu-K$\alpha$ radiation. Morphological and structural analyses were performed by scanning electron microscope (SEM, HITACHI S4800)

with energy dispersive spectroscopy (EDS) and field emission transmission electron microscope (FE-TEM, FEI Tecnai F30). X-ray photoelectron spectroscopy (XPS) measurements were carried out by PHI 5000 VersaProbe II. The XPS in-depth tests for SEI (20 nm) and CEI (4 nm) analysis were conducted after cycling Li||NCM811 full cells 20 times at 0.5C and 25 °C. The SEI component was collected by time-off light secondary ion mass spectrometry (ToF-SIMS, PHI nanoTOF II, 30 keV, 2 nA) in a 200 μm (length) × 200 μm (width) ×50 nm (thickness) region. The $^1$H, $^7$Li, and $^{19}$F nuclear magnetic resonance (NMR) were performed with a Bruker 600 MHz AVANCE III spectrometer. The atomic force microscopy-nano-infrared spectroscopy (AFM-nano-IR) measurements were undertaken with Bruker Anasys nanoIR2-fs. The Fourier transform infrared (FTIR) spectra were executed by VERTEX 70 spectrometer in attenuated total reflection (ATR) mode. The Raman spectra were obtained by LabRAM HR Evolution. The roughness and Young's modulus of the SEI were measured by AFM (Bruker Dimension Ico). The CEI images on the cycled NCM811 were collected by FEI Tecnai F30. Thermogravimetric analysis (TGA) was performed by Netzsch STA 449F3 thermal analyzer from room temperature to 600 °C at heating rate of 10 °C min$^{-1}$ in N$_2$ atmosphere. The BDS measurement was conducted on Novocontrol Concept 80 broadband dielectric spectrometer (Montabaur, Germany) with temperature control. The applied voltage was 1.0 V$_{rms}$ with frequency ranging from 10$^{-1}$ to 10$^7$ Hz and temperature from −50 °C to 100 °C.

### Electrochemical measurements

The ionic conductivities were tested by electrochemical impedance spectroscopy (EIS) from 1 MHz to 0.1 Hz with a 20 mV AC oscillation on a VMP3 multichannel electrochemical station (Bio-Logic Science Instruments, France). Prior to the EIS measurements, the cells were kept at each test temperature for 1 h to reach the thermal equilibrium. The ionic conductivities ($\sigma$) were calculated following Eq. (1):

$$\sigma = \frac{L}{RS} \tag{1}$$

where $L$ is the thickness of electrolytes, $R$ is obtained by EIS measurement with electrolytes sandwiched between two stainless plates of steel, and $S$ is the area of stainless steel (SS, diameter: 15.8 mm). The activation energy was calculated from the Arrhenius Eq. (2):

$$\sigma = \sigma_0 \exp\left(-\frac{E_a}{RT}\right) \tag{2}$$

where $\sigma_0$ and $E_a$ is the pre-exponential factor and the activation energy of ions transportation, respectively.

The electronic conductivities were tested in symmetric cells with stainless-steel spaces as electrodes and conducted via Chronoamperometry (CA) tests with an applied voltage of 500 mV.

The linear sweep voltammetry (LSV) curves were collected at 0-6 V (vs. Li/Li$^+$) with scan rate of 0.1 mV s$^{-1}$ using VMP3 multichannel electrochemical station. The cyclic voltammetry (CV) curves of Li||NCM811 full cells were obtained at a scanning rate of 0.05 mV s$^{-1}$ at 2.8–4.3 V (vs. Li/Li$^+$). Galvanostatic charge/discharge tests of cells were performed on LAND CT2001A and Neware battery test systems. The cycled cells were transferred into a glovebox and dissembled for characterizations. The cycled Li metal anode (lithium foil, thickness: 450 μm, diameter: 15.6 mm) and NCM811 cathode (diameter: 12 mm) were transferred into a chamber with a sealed Ar-filled vessel for SEM and XPS examinations.

The determination of $\beta$-phase PVDF relative content can be done using Lambert-Beer law by using Eq. (3):

$$F(\beta) = \frac{A_\beta}{\left(\frac{K_\beta}{K_\alpha}\right)A_\alpha + A_\beta} \times 100\% \tag{3}$$

where $F(\beta)$ gives the content of $\beta$ phase in the respective nano-composite film, $K_\beta = 7.7 \times 10^4 \, cm^2 \, mol^{-1}$ is the absorption coefficient at $840 \, cm^{-1}$ and $K_\alpha = 6.1 \times 10^4 \, cm^2 \, mol^{-1}$ is the absorption coefficient at $762 \, cm^{-1}$. Absorption at $762 \, cm^{-1}$ is denoted by $A_\alpha$ and for $840 \, cm^{-1}$ is denoted by $A_\beta$.

The critical current density (CCD) tests are performed using the Li‖Li symmetric cells using the time control and capacity control methods. For the time control method, the Li‖Li symmetric cells were cycled for 5 times at current density of $0.05 \, mA \, cm^{-2}$ to form the SEI, which were then cycled from $0.1 \, mA \, cm^{-2}$ with current density increasing $0.1 \, mA \, cm^{-2}$ per cycle, while the charge/discharge time remaining fixed for 1 h. For the capacity control method, the Li‖Li cells were cycled for five times at current density of $0.05 \, mA \, cm^{-2}$ to form the SEI, which were then cycled from $0.1 \, mA \, cm^{-2}$ with current density increasing $0.1 \, mA \, cm^{-2}$ per cycle, while the charge/discharge capacity remaining fixed for $0.2 \, mAh \, cm^{-2}$.

## Fabrication of cells

The cathode was prepared by mixing $LiNi_{0.8}Co_{0.1}Mn_{0.1}O_2$ (NCM811, $D_{50}$ = 3.77 μm), Super P (particle size: 40–50 nm, 99.5%), PVDF 5130 binder ($M_w$ = 1,200,000 Da, particle size: 100 μm, 99.5%) and LiTFSI (Sigma-Aldrich, 99.95%) in a weight ratio of 75:10:10:5 (0.75 g:0.1 g:0.1 g:0.05 g) in N-methyl-2-pyrrolidone (NMP, 99.9%, Aladdin), followed by casting the slurry on an Al foil (thickness: 10 μm, 99.9%). After drying at 80 °C for 12 h, the cathode was prepared with NCM811 mass loading around $2 \, mg \, cm^{-2}$. The CR2032 (spacer: 15.8 × 1 mm, spring: 15.4 × 1.1 mm) solid-state cells were assembled in an Ar-filled glovebox. The CR2032 liquid cells were assembled using Celgard 2500 separator and 50 μL l M $LiPF_6$ in EC/DMC/EMC (1:1:1) organic electrolyte. For Li‖NCM811 pouch cell-1, single-side coated cathode (mass loading: $2 \, mg \, cm^{-2}$, 4 cm × 4.5 cm) and one double-coated anode (4.5 cm × 5 cm, 20 μm Li for each side) were stacked one by one and separated by PVMS-15 electrolyte (5 cm × 5.5 cm). For Li‖NCM811 pouch cell-2, single-side coated cathode (mass loading: $8 \, mg \, cm^{-2}$, 3 cm × 3.5 cm) and single-coated anode (4.5 cm × 5 cm, 50 μm Li) were stacked one by one and separated by PVMS-15 electrolyte (5 cm × 5.5 cm). Li‖Cu (Cu foil, thickness: 10 μm, 99.9%) cells were assembled to test CE and Li deposition morphology.

## Surface structure construction

The equilibrium lattice constant of $Li_2Se$, $Li_2O$, $Li_2CO_3$, and LiOH unit cell was optimized using a Gamma-center grid for Brillouin zone sampling with K-mesh density equal to $0.19 \, Å^{-1}$ (Supplementary Table 3). We then cleave surface models with periodicity in the $x$ and $y$ directions based on the original cells. A vacuum spacing of 15 Å was introduced in the surface diffusion calculations to avoid the artificial interaction between periodic duplicates. The surface orientation selected for diffusion energy barrier calculation is the most stable surface in the TEM experiment, and the results of the surface energy calculation also confirm this (Supplementary Table 4). The surface energy is defined as following Eq. (4):

$$\gamma_{surface} = \frac{1}{A}\left(E_{surf}^N - N \times E_{bulk}\right) \tag{4}$$

Where $E_{surf}^N$ is the total energy of the optimized slab model containing N atoms, $E_{bulk}$ is the bulk energy of a perfect crystal unit cell per atom, A is the surface area of the slab model. Finally, the $Li_2Se$ (111) surface

model contains 54 Li and 27 Se atoms, the $Li_2O$ (111) surface model contains 32 Li and 16 O atoms, the $Li_2CO_3$ (002) surface model contains 24 Li, 12 C atoms, and 36 O atoms, and LiOH (001) surface model contains 36 Li, 36 O atoms, and 36 H atoms. All of the atomic configurations were visualized using VESTA.

## Adsorption energy calculations

The geometry optimization was calculated by minimizing the energy with Conjugate gradient (CG) quench using the Hellmann–Feynman forces. The kinetic energy cutoff is set at 400 eV for the plane-wave expansion. A Monkhorst–Pack scheme of $2 \times 2 \times 1$ with ($MoSe_2$ adsorption series system) k-point mesh were used for Brillouin zone sampling. The convergence criterions for total energy and force were set at $10^{-5} \, eV$ and $10^{-2} \, eV \, Å^{-1}$, respectively.

## Diffusion energy barrier calculation details

The climbing image-nudged elastic band (CI-NEB) method was employed to determine the energy barrier of the Li ion diffusion[60]. The initial guess of the transition states was generated by the image-dependent pair potential (IDPP) method[61]. By convergence tests of potential energy, the kinetic energy cutoff is set at 520 eV for the plane-wave expansion. The VASP.5.4.4 potential set of Li (PAW_PBE Li_sv 10Sep2004), Se (PAW_PBE Se 06Sep2000), O (PAW_PBE O 08Apr2002), C (PAW_PBE C 08Apr2002), H (PAW_PBE H 15Jun2001) was used for elemental valence electron configurations in diffusion energy barrier calculations. In CI-NEB calculations, the bottom 4–6 stoichiometric layers were fixed while the top two layers were allowed to relax, according to different systems. We utilized a $2 \times 2 \times 1$ Gamma-center k-point mesh for the $Li_2Se$ (111), $Li_2O$ (111), $Li_2CO_3$ (002), and LiOH (001) surface diffusions for Brillouin zone sampling. The dispersion correction D3 was included in the DFT calculation by using the zero damping approach. The Gaussian smearing with a parameter σ of 0.1 eV was used for electronic iterations after testing the effect of σ on the electron entropy, and the convergence criteria for total energy and force were set at $10^{-7} \, eV$ and $10^{-2} \, eV \, Å^{-1}$, respectively. The calculated transition state structures were further confirmed by vibrational analysis to ensure only one imaginary normal mode of vibration leading to the final state considered existing.

## Data availability

The authors declare that the data supporting the findings of this study are available within the paper and its Supplementary Information files. Should any raw data files be needed in another format, they are available from the corresponding author upon reasonable request. Source data are provided with this paper.

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

## Acknowledgements

We acknowledge financial support from the National Key R&D Program of China (2018YFA0209600), the National Natural Science Foundation of China (22022813 and 21878268), and the Leading Innovative and Entrepreneur Team Introduction Program of Zhejiang (2019R01006), the Key R&D Program of Zhejiang Province (2019C01155), and the Fundamental Research Funds for China Central Universities (DUT22LAB608).

We thank the Center of Cryo-Electron Microscopy (CCEM), Zhejiang University for the technical assistance on Cryo-EM. We thank Mrs. Na Zheng at State Key Laboratory of Chemical Engineering in Zhejiang University for performing SEM. We thank Mrs. Jing He at State Key Laboratory of Chemical Engineering in Zhejiang University for performing atomic force microscopy-nano-infrared spectroscopy (AFM-nano-IR).

## Author contributions

Y.L. conceived the idea and supervised the project. Q.W. and Y.L. designed the experiments. Q.W. performed the experiments with the help of M.F., S.J., S.L., S.Z., Z.S., S.M., J.M., J.Z., Y.T., K.S., and J.L. M.F. and S.J. performed the calculations. All authors discussed the results in the manuscript. Q.W., W.H., Y.H., and Y.L. wrote the initial paper, which was approved by all the authors.

## Competing interests

The authors declare no competing interests.
