## [Peer Review File · Nature Communications]

Phase regulation enabling dense polymer-based composite electrolytes for solid-state lithium metal batteriesREVIEWER COMMENTS

Reviewer #1 (Remarks to the Author):

This manuscript describes addition of MoSe₂ sheets to a PVDF electrolyte with trace solvent. The addition of the secondary phase appears to change the proportion of PVDF phases, yielding a denser, higher ionic conductivity electrolyte. What is more, the addition of MoSe₂ appears to modify the interphase formed against a Li metal anode, leading to a lower resistance interphase. The authors suggest that this is responsible for the cycling performance of the modified electrolyte, which is clearly improved when compared to a PVDF electrolyte with no added MoSe₂.

There are many good points to this work, in particular the improved conductivity, activation energy and cycling performance of the electrolyte. However, I cannot recommend that this manuscript is published in Nature Communications as there are some errors in data analysis which could potentially compromise some of the conclusions. More detailed comments and questions follow:

1. Line 132, the authors say inhomogeneous DMF “always induces uneven Li depositions and rapid Li dendrites growth”. Can authors provide evidence for this? It seems to be pertinent to key point that the PVMS-15 is able to achieve higher critical current.
2. At the current scale the FTIR Fig 2a does not show clear differences and manuscript is unclear on how the ratio of phases have been calculated from this.
3. Fittings of FTIR and Raman are reported with high accuracy. What are the errors on these fittings?
4. High CCD is a key claim of the manuscript, however it is not clear how this experiment was performed as insufficient experimental details have been given. The graph does not make sense. Time is on the x-axis and the current has presumably been increased in steps as is customary with a CCD test (10.1002/adfm.202009925). If this is the case, why does each cycle appear to have taken the same time? If capacity is fixed, cycle time should decrease with increasing current density.
5. This manuscript should not be published with the current analysis of XPS data. There are a number of serious issues, some of which I give here:
 - i. The fittings of S 2p (Fig 5c and Supp. Fig 34), Se 3d (Supp. Fig 30a) and Mo 3d (Supp. Fig 30b) are not physical. These have all been fitted as singlet peaks, but p-orbital peaks must be doublets in a 2:1 ratio and d-orbital peaks must be doublets with a 3:2 ratio. Fwhm should remain constant (although some variation may be required when fitting Mo).
 - ii. C 1s fittings are at odds with what is commonly observed in literature. It is uncommon to see C=O without C-O and other similar species (Fig 5a), and in Fig 5g C-C and C-H are at very different binding energies when they are indistinguishable in position.
 - iii. Quantification in Figs 5d and e are based on incorrect fittings. Additionally, some elements quantified e.g. lithium and oxygen are not shown in the main manuscript or supplementary.
 - iv. Can the authors comment on the choice to show S2p in case of Li anode interphase (c), but N 1s in case of cathode interphase (i)?
 - v. Authors state “the obviously weaker XPS intensities using PVMS-15 electrolyte suggest the suppressed interfacial side reaction.” This is not necessarily the case for comparing absolute intensity of peaks between two different samples.
6. The manuscript is challenging to read and the English needs to be improved. Sometimes incorrect terms have been used e.g. page 11 “interface” is used instead of interphase.
7. Experimental methods are not sufficiently detailed for this work to be reproduced.

Reviewer #2 (Remarks to the Author):

In this manuscript, Lu and coworkers present a study on the preparation of a composite polymer electrolyte and its application for solid-state batteries.

In particular, the authors combine PVDF with molybdenum diselenide particles and characterize such a composite with many methods, both experimental and theoretical. As an experimentalist myself, I can only provide comments on the experimental results.

One aspect that is unclear to me is why the authors decided to incorporate molybdenum diselenide (MoSe₂) for the electrolyte composites (besides it serving as a source for some special type of SEI or CEI component). It is well known that MoSe₂ is a semiconducting material, and if I am not mistaken, an n-type material. As such, MoSe₂ is capable of transporting electrons. When combining MoSe₂ with the PVDF, Li salt and DMF, the composites are effectively behaving as a mixed ionic-electronic conductor. Such properties are quite undesired in an electrolyte for a battery. Because of the electronically conducting properties of the MoSe₂, I believe that the improved conductivity of the composite polymers (reported in Figure S14) is due to the fact that both ions and electrons can move in the membrane (net increase of charge carriers), and not because the ionic conductivity is significantly increased. To demonstrate that the ionic conductivity of the composite is improved, the authors should consider measuring the partial ionic and electronic conductivities of the composite polymers independently and demonstrate the effect of MoSe₂ on both. Such measurements are possible using symmetrical cells with ion-blocking electrodes (for partial electronic conductivity) and reversible electrodes and solid electrolyte blocking layers (for partial ionic conductivity). A thorough description of such measurements using impedance spectroscopy for characterizing NMC-solid electrolyte composites was reported by Minnmann et al. (J Electrochem Soc 2021, 168, 040537). As a final note here, it is unclear how much solvent (DMF) is present in the composite. Can the authors provide a weight or volume percentage in the experimental description of the sample preparation?

Despite the limited knowledge on the transport processes taking place in the composite polymer electrolyte, incorporating such a material in cells seems to have some positive effect in the cycling performance (Figure 4). However, I think it is unfair to the readers to present such small plots, and also to have the axis for the Coulombic Efficiencies range from 0 to 100. At such a scale it is very difficult to ascertain what is the “real” behavior of the cells. The comparison to other reports in Figure 4g is unreasonable because the authors do not consider difference of CAM loadings, which are known to have a strong effect on cycle number and rate capabilities in SSBs. Along these lines, I consider a CAM loading of ca. 2 mg/cm² not very competitive to other reports in SSBs. For example, sulfide- and halide-solid electrolyte based SSBs (press-cells) with CAM loadings of 10 – 20 mg/cm² CAM loadings have been reported by the Nazar and Janek groups recently. At the low loadings reported here, it is difficult to gauge the transferability of these results for higher loadings and/or thicker cathodes.

Reviewer #3 (Remarks to the Author):

The authors report a PVDF-based gel electrolyte with MoSe₂ fillers. The MoSe₂ sheets are able to interact with PVDF molecules and alter the solvation structure of Li⁺. As a result, the electrolyte shows improved density, ionic conductivity, electrochemical stability, and

mechanical properties. The authors have performed thorough characterizations and show decent electrochemical performances. I suggest that this article can be published after minor revisions. The following questions and advices are for the authors:

1. Please label the molecules properly in Fig. 1b.
2. Please distinguish "interface" and "interphase".
3. The SEM images of PVMS-15 and PVDF should be compared at the same magnification.
4. In Fig. 1e, it seems that the electrolyte is not homogeneous and there is a phase separation. The PVDF phase looks still a bit porous. Please provide more SEM images at different areas with higher magnifications.
5. 15% MoSe₂ in the polymer should be discontinuous in the electrolyte and tends to aggregate according to Fig. 1e. Do you really have a SEI with a uniform chemical composition? The Li₂Se and Mo content in the SEI should vary dramatically at different spots.
6. Please add a scale bar in Supplementary Fig. 31.
7. The authors claim that the Li⁺-DMF coordination is strengthened in the electrolyte, which should also facilitate the decomposition of DMF. However, the authors claim the opposite. How does the Li₂Se-containing SEI suppress DMF decomposition?
8. What is the ion-conductor in the NMC811 cathode?
9. Please specify the loading and/or areal capacity of the cathode and the thickness of the Li anode in the main text.
10. Fig. 4g is not a fair comparison. Please add an axis of areal capacity and change the C-rate to current density in the graph.

Response to Referees' Comments

Dear Editors and Referees:

Thank you very much for your precious time in reviewing our manuscript (Manuscript ID: NCOMMS-22-53461) and providing quite valuable comments. We have read your comments carefully and tried our best to revise manuscript accordingly. We think all comments are quite helpful and important to raise the quality of the revised manuscript. All comments have been point-by-point replied as below and the revised parts have been highlighted in the revised manuscript with **YELLOW** background. We sincerely look forward to the reviewer's approval and expect this revised manuscript can reach the high standard of Nature Communications.

Reviewers' comments:

Reviewer #1 (Remarks to the Author):

This manuscript describes addition of MoSe₂ sheets to a PVDF electrolyte with trace solvent. The addition of the secondary phase appears to change the proportion of PVDF phases, yielding a denser, higher ionic conductivity electrolyte. What is more, the addition of MoSe₂ appears to modify the interphase formed against a Li metal anode, leading to a lower resistance interphase. The authors suggest that this is responsible for the cycling performance of the modified electrolyte, which is clearly improved when compared to a PVDF electrolyte with no added MoSe₂.

There are many good points to this work, in particular the improved conductivity, activation energy and cycling performance of the electrolyte. However, I cannot recommend that this manuscript is published in Nature Communications as there are some errors in data analysis which could potentially compromise some of the conclusions. More detailed comments and questions follow:

Reply: Thank you very much for your quite significant comment. We are very sorry to make some mistakes on the data analysis and processing, especially on peaks fitting and data comparison of the XPS data and definition of the error bars. In the revised manuscript, we have carefully recollected and reasonably analyzed the data. We believe the quality of this revised version have been greatly improved, including the characterizations of the solid electrolyte interphase (SEI)/cathode electrolyte interphase (CEI), the practical performance of solid-state Li||NCM811 batteries and the English writing of texts. We sincerely look forward to the reviewer's approval for this revised manuscript owing to the following novel perspectives and significance in this work.

First, we adopt an innovative phase regulation strategy based on asymmetric adsorption interactions to disrupt the symmetry of PVDF chains, which results in the formation of all trans conformation of the PVDF and dense structure of the composite electrolyte by coupling MoSe₂ sheets. This is the first report focusing on the densification of PVDF-based electrolytes. More importantly, many associated benefits including improve the ionic conductivity and widen the electrochemical stability window of the electrolyte, suppress the side reactions with electrodes, increase the coulombic efficiency could be achieved by this ingenious design. This work can address several key issues of PVDF-based electrolytes and provide an encouraging path towards their large-scale production in the practical application.

Second, for the first time, we reveal that the formed solvation structures by residual N, N-dimethylformamide (DMF) solvent and Li salt in PVDF-based electrolytes are quite similar with the widely reported liquid high-concentrated electrolyte (HCE) while the formed SEI/CEI possesses unstable feature, which is quite important to understand the fundamentals of microscope properties and boost the macroscopic performance for PVDF-based electrolytes. Here, we found that tailoring the inherent solvation structures and modifying the interphase component is quite effective to improve the ion transport capability and suppress the side reactions, which may open a new research direction.

Third, we believe this is a time-efficient and important work. During the revisions of our manuscript, our proposed concepts could be accepted by peer researchers. For instance, Shi et al. reported that dielectric $\text{BaTiO}_3\text{-Li}_{0.33}\text{La}_{0.56}\text{TiO}_{3-x}$ filler can also increase the dielectric constant (ϵ_r) of PVDF-based electrolytes to promote the dissociation of Li salt and increase ionic conductivity (*Nat. Nanotechnol.* 2023, <https://doi.org/10.1038/s41565-023-01341-2>). Then, Huang et al. found that the all trans conformation (β phase) PVDF could assist to form uniform ion transport channels in the PVDF electrolyte (*Adv. Energy Mater.* 2023, 2203888). Later, Liu et al. reported a novel polymer P(VDF-TrFE-CTFE) with higher polarity than PVDF, which can also promote the dissociation of Li salt (*Angew. Chem. Int. Ed.* 2023, e20230243). Therefore, the concept and strategy of our manuscript is useful to this field.

In all, we believe our work is significant and the electrochemical performance of our developed batteries is competitive. We sincerely look forward to the reviewer's approval and expect this revised manuscript can reach the high standard of *Nature Communications*.

1. Line 132, the authors say inhomogeneous DMF “always induces uneven Li depositions and rapid Li dendrites growth”. Can authors provide evidence for this? It seems to be pertinent to key point that the PVMS-15 is able to achieve higher critical current.

Reply: Thank you very much for your quite significant comment.

As revealed by Fig. 1f, the PVDF electrolyte exhibits a porous structure due to the phase separation between the polymer and solvent during the preparation process, and the DMF solvent aggregates around the PVDF spherulites (Supplementary Fig. 9 and Fig. 1i). Thus, the inhomogeneous DMF could induce uneven Li deposition and rapid Li dendrite growth. This point has been evidenced by previous reports. For instance, “the porous structure of the PVDF matrix induces heterogeneous ions flux, which results in uneven Li depositions and rapid dendrites growth” was reported by Yang et al (*Angew.*

Chem. Int. Ed. 2021, 60, 24668-24675) and “the PVDF-HFP matrix in SPEs is composed of micro-sized spherical particles with voids forming between the particles. These voids may reduce the mechanical strength and cause uneven distribution of lithium-ion flux, resulting in uncontrolled Li dendrite growth” was reported by Zhai et al (*Adv. Energy Mater.* 2023, 12, 2200967).

To further confirm this, we directly observed the Li deposition morphology in Li|Cu cells and tested the maximum Li deposition capacity in Li|Li cells (Supplementary Note 1). First, the plating morphologies of Li (1 mAh cm^{-2}) on Cu substrates were examined. As shown in Supplementary Fig. 10, the Li|PVDF|Cu cell present a highly loose and irregular deposition structure with many small Li particles, demonstrating the uneven Li deposition with the PVDF electrolyte. Second, we assembled Li|PVDF|Li cell to test the maximum Li deposition capacity. The Li|PVDF|Li cell was cycled 5 times under current density of 0.1 mA cm^{-2} to form the SEI, which was then operated by continuously charge (plating Li from one side to the other side) until the battery presents a short circuit. As shown in Supplementary Fig. 11, the maximum Li deposition capacity is approximately 1 mAh cm^{-2} . Given this, we can conclude that the uneven Li deposition and limited capacity always induce rapid Li dendrite growth and restrict the performance of the solid-state batteries under high loadings. In contrast, the maximum Li deposition capacity is as high as 2.6 mAh cm^{-2} with the dense PVMS-15 electrolyte (Supplementary Fig. 30). After plating 1 mAh cm^{-2} Li on the Cu foil, dense and connected Li particles can be observed with the PVMS-15 electrolyte (Supplementary Fig. 32).

Supplementary Figure 10. SEM images of the Li deposition obtained by plating 1 mAh cm⁻² Li on Cu substrate at 0.1 mA cm⁻² in Li||Cu cells using PVDF electrolyte. a 500×. b 2000×.

Supplementary Figure 11. Maximum Li deposition capacity tested in Li||Li cells at 0.1 mA cm⁻² using PVDF electrolyte.

Supplementary Figure 30. Maximum Li deposition capacity tested in Li||Li cells at 0.1 mA cm⁻² using PVMS-15 electrolyte.

Supplementary Figure 32. SEM images of the Li deposition obtained by plating 1 mAh cm⁻² Li on Cu substrate at 0.1 mA cm⁻² in Li||Cu cells using PVMS-15 electrolyte. a 500×. b 2000×.

2. At the current scale the FTIR Fig 2a does not show clear differences and manuscript is unclear on how the ratio of phases have been calculated from this.

Reply: Thank you very much for your significant comment. We have revised the Fig. 2a and provide details on calculation of β -phase PVDF in Methods and Supplementary Fig. 18-19 and Table 1.

As shown in Fig. 2a, the peaks corresponding to the β -phase PVDF are marked at 840 and 1388 cm⁻¹, and the peaks corresponding to 762 and 1360 cm⁻¹ are related to the α -phase PVDF. The average content of the β -phase is calculated to be 39% using the Lambert-beer law for the PVDF electrolyte (Equation 3 in Method, details in Supplementary Fig. 18 and Table 1). In contrast, with increasing amounts of MSs, the ratio of β -phase PVDF increases to 52%, 64% and 77% for the PVMS-10, PVMS-15 and PVMS-20 electrolytes, respectively. (line 148-154, page 7)

Fig. 2a FTIR spectra of the electrolytes

Supplementary Figure 18. β-phase PVDF content of the electrolytes.

The determination of β-phase PVDF relative content can be done using Lambert-Beer law by using equation (3):

$$F(\beta) = \frac{A_{\beta}}{\left(\frac{K_{\beta}}{K_{\alpha}}\right)A_{\alpha} + A_{\beta}} \times 100\% \quad (3)$$

where $F(\beta)$ gives the content of β phase in the respective nanocomposite film, $K_{\beta} = 7.7 \times 10^4 \text{ cm}^2 \text{ mol}^{-1}$ is the absorption coefficient at 840 cm^{-1} and $K_{\alpha} = 6.1 \times 10^4 \text{ cm}^2 \text{ mol}^{-1}$ is the absorption coefficient at 762 cm^{-1} . Absorption at 762 cm^{-1} is denoted by A_{α} and for 840 cm^{-1} is denoted by A_{β} . (line 418-422, page 21)

Supplementary Table 1. Calculation details of β-phase PVDF in the electrolytes.

Electrolyte	Average A_{β}	Average A_{α}	Average β-phase PVDF content (%)
PVDF	0.1598	0.1891	39
PVMS-10	0.2014	0.1430	52
PVMS-15	0.2422	0.1057	64
PVMS-20	0.2891	0.0661	77

3. Fittings of FTIR and Raman are reported with high accuracy. What are the errors on these fittings?

Reply: Thank you very much for your significant comment. We have added the error bars on the FTIR (Supplementary Fig. 18 and Table 1) and Raman (Fig. 2e) data.

Fig. 2e Raman spectra results of the PVDF and PVMS-15 electrolytes.

Supplementary Figure 18. β -phase PVDF content of the electrolytes.

Supplementary Table 1. Calculation details of β -phase PVDF in the electrolytes.

Electrolyte	Average A_β	Average A_α	Average β -phase PVDF content (%)
PVDF	0.1598	0.1891	39
PVMS-10	0.2014	0.1430	52
PVMS-15	0.2422	0.1057	64
PVMS-20	0.2891	0.0661	77

4. High CCD is a key claim of the manuscript, however it is not clear how this experiment was performed as insufficient experimental details have been given. The graph does not make sense. Time is on the x-axis and the current has presumably been increased in steps as is customary with a CCD test (10.1002/adfm.202009925). If this is the case, why does each cycle appear to have taken the same time? If capacity is fixed,

cycle time should decrease with increasing current density.

Reply: Thank you very much for your significant comment. We have added the experimental details on the CCD test and tested the CCD using capacity control and time control methods (*Adv. Funct. Mater.* 2021. 31, 2009925). The time control method helps to assess the endurance with simultaneously increased capacity and current density, while the capacity control protocol is effective to evaluate the influence of capacity during cycling.

The critical current density (CCD) tests are performed using the Li||Li symmetric cells using the time control and capacity control methods. For the time control method, the Li||Li symmetric cells were cycled for 5 times at current density of 0.05 mA cm⁻² to form the SEI, which were then cycled from 0.1 mA cm⁻² with current density increasing 0.1 mA cm⁻² per cycle, while the charge/discharge time remaining fixed for 1 h. For the capacity control method, the Li||Li cells were cycled for 5 times at current density of 0.05 mA cm⁻² to form the SEI, which were then cycled from 0.1 mA cm⁻² with current density increasing 0.1 mA cm⁻² per cycle, while the charge/discharge capacity remaining fixed for 0.2 mAh cm⁻². (line 423-429, page 21)

We then cycled Li||Li cells to assess the properties of the Li₂Se-containing SEI. As shown in Fig. 3n and Supplementary Fig. 28, the Li|PVMS-15|Li cell enables critical current densities (CCDs) of 2.3 mA cm⁻² and 8.3 mA cm⁻² using the time control and capacity control methods⁵⁵, respectively, which are much higher than those of the Li|PVDF|Li cell (0.8 mA cm⁻² and 1.6 mA cm⁻²), indicating that Li dendrite growth can be effectively suppressed. (line 247-251, page 13)

Fig. 3n CCD test curves with time control method.

Supplementary Figure 28. CCD tests using capacity control method.

5. This manuscript should not be published with the current analysis of XPS data. There are a number of serious issues, some of which I give here:

Reply: Thank you very much for your significant comment. We are very sorry to make some mistakes on the data analysis and processing, especially on peaks fitting and data comparison of the XPS data. In the revised manuscript, we have carefully recollected and reasonably analyzed many data.

i. The fittings of S 2p (Fig 5c and Supp. Fig 34), Se 3d (Supp. Fig 30a) and Mo 3d (Supp. Fig 30b) are not physical. These have all been fitted as singlet peaks, but p-orbital peaks must be doublets in a 2:1 ratio and d-orbital peaks must be doublets with a 3:2 ratio. Fwhm should remain constant (although some variation may be required when fitting Mo).

Reply: Thank you very much for your significant comment. We have revised the fittings of S 2p, Se 3d and Mo 3d (Supplementary Fig. 43 and 48).

ii. C 1s fittings are at odds with what is commonly observed in literature. It is

uncommon to see C=O without C-O and other similar species (Fig 5a), and in Fig 5g C-C and C-H are at very different binding energies when they are indistinguishable in position.

Reply: Thank you very much for your significant comment. We have revised the fittings of C 1s (Supplementary Fig. 43 and 48).

iii. Quantification in Figs 5d and e are based on incorrect fittings. Additionally, some elements quantified e.g. lithium and oxygen are not shown in the main manuscript or supplementary.

Reply: Thank you very much for your significant comment. We have revised the fittings and added XPS results of Li and O (Supplementary Fig. 43 and 48).

iv. Can the authors comment on the choice to show S2p in case of Li anode interphase (c), but N 1s in case of cathode interphase (i)?

Reply: Thank you very much for your significant comment. In case of SEI/CEI, the S 2p, F 1s and N 1s spectra is related to the decomposition of FSI, while the C 1s spectra is mainly ascribed to the decomposition of solvent. We are very sorry for our incorrect data analysis and processing. In the revised manuscript, we have recollected the XPS results for CEI/SEI and provided depth profiles of atomic ratios at different sputter time (Fig. 5a, Supplementary Fig. 43, 44, 48 and 49).

v. Authors state “the obviously weaker XPS intensities using PVMS-15 electrolyte suggest the suppressed interfacial side reaction.” This is not necessarily the case for comparing absolute intensity of peaks between two different samples.

Reply: Thank you very much for your significant comment. We are very sorry for our incorrect data analysis and processing. In the revised manuscript, we have recollected the XPS results for CEI/SEI and provided depth profiles of atomic ratios at different sputter time to show the influence of PVMS-15 electrolyte on the interfacial chemistry. (Fig. 5a, Supplementary Fig. 43, 44, 48 and 49)

The X-ray photoelectron spectroscopy (XPS) results show that the SEI mainly consists of C-containing organic species and inorganic species including Li_2CO_3 , Li_2O , LiF and

sulfur compounds (Supplementary Fig. 43), which are attributed to the decomposition of DMF solvent and FSI²⁹, respectively. Li₂Se and Mo can be clearly detected, demonstrating that reactions between MSs and Li metal can occur. As shown in Fig. 5a and Supplementary Fig. 44, the C and O atomic contents are decreased with the PVMS-15 electrolyte, demonstrating that DMF decomposition is effectively mitigated. The Se and Mo contents at different sputtering times remain constant at approximately 4.68% and 2.53%, respectively, indicating a uniform distribution of Li₂Se and Mo. (line 318-326, page 17)

In terms of the cathode electrolyte interphase (CEI) for cycled NCM811, the lower C and O atomic contents demonstrate suppressed electrolyte oxidation during cycling (Supplementary Fig. 48-49). (line 341-342, page 18)

Fig. 5a Atomic ratios of the SEI at different sputter time with the PVMS-15 electrolyte.

Supplementary Figure 44. Atomic ratios of the SEI at different sputter time with the PVDF electrolyte.

Supplementary Figure 49. Atomic ratios of the CEI at different sputter time with the PVMS-15 (a) and PVDF (b) electrolytes.

Supplementary Figure 43. XPS results of SEI using PVDF and PVMS-15 electrolytes.

Supplementary Figure 48. XPS results of CEI using PVDF and PVMS-15 electrolytes.

6. The manuscript is challenging to read and the English needs to be improved. Sometimes incorrect terms have been used e.g. page 11 “interface” is used instead of interphase.

Reply: Thank you very much for your significant comment. We have revised incorrect terms including the solid electrolyte interphase (SEI) and cathode electrolyte interphase (CEI). The revised manuscript was edited by Springer Nature Author Services for grammar, phrasing, and punctuation. In addition, many edits were made to further improve the flow and readability of the text.

7. Experimental methods are not sufficiently detailed for this work to be reproduced.

Reply: Thank you very much for your significant comment. We have added details of experimental methods. (line 370-443, page 20-21)

Reviewer #2 (Remarks to the Author):

In this manuscript, Lu and coworkers present a study on the preparation of a composite polymer electrolyte and its application for solid-state batteries. In particular, the authors combine PVDF with molybdenum diselenide particles and characterize such a composite with many methods, both experimental and theoretical. As an experimentalist myself, I can only provide comments on the experimental results.

Reply: Thank you very much for the valuable comments on our work. We have carefully

revised our manuscript point by point according to your valuable comments below.

One aspect that is unclear to me is why the authors decided to incorporate molybdenum diselenide (MoSe_2) for the electrolyte composites (besides it serving as a source for some special type of SEI or CEI component). It is well known that MoSe_2 is a semiconducting material, and if I am not mistaken, an n-type material. As such, MoSe_2 is capable of transporting electrons. When combining MoSe_2 with the PVDF, Li salt and DMF, the composites are effectively behaving as a mixed ionic-electronic conductor. Such properties are quite undesired in an electrolyte for a battery. Because of the electronically conducting properties of the MoSe_2 , I believe that the improved conductivity of the composite polymers (reported in Figure S14) is due to the fact that both ions and electrons can move in the membrane (net increase of charge carriers), and not because the ionic conductivity is significantly increased. To demonstrate that the ionic conductivity of the composite is improved, the authors should consider measuring the partial ionic and electronic conductivities of the composite polymers independently and demonstrate the effect of MoSe_2 on both. Such measurements are possible using symmetrical cells with ion-blocking electrodes (for partial electronic conductivity) and reversible electrodes and solid electrolyte blocking layers (for partial ionic conductivity). A thorough description of such measurements using impedance spectroscopy for characterizing NMC-solid electrolyte composites was reported by Minnmann et al. (J Electrochem Soc 2021, 168, 040537).

Reply: Thank you very much for your significant comment. Our design proposal is that regulating the phase of PVDF polymer chains to obtain a dense electrolyte film toward practical application. Compared with other conformations of PVDF such as α and γ , the β -phase PVDF presents the highest polarity and dielectric properties, which could promote the Li salt dissociation and enhance ionic conductivity of the electrolytes. (Nat. Nanotechnol. 2023. <https://doi.org/10.1038/s41565-023-01341-2>; Energy Environ. Sci. 2021, 14, 6021-6029). Thus, we try to enhance the ratio of β -phase PVDF via a phase transformation strategy by coupling with fillers that can interact with the dipole moment of PVDF. Interesting, we find that that the MoSe_2 with broken symmetry may interacts

with PVDF, which motivated us to add MoSe₂ as fillers in the PVDF electrolyte. The results show that there is a positive correlation between the ratio of β -phase PVDF and ϵ_r after the addition of MoSe₂ (Fig. 2d). As expected, we obtained a dense PVMS-15 electrolyte with enhanced β -phase PVDF and superior ionic conductivity. In addition, many associated benefits including widening the electrochemical stability window of the electrolyte, suppressing the side reactions with electrodes, and increasing the coulombic efficiency could be achieved by this ingenious design. This is the first report focusing on the densification of PVDF-based electrolytes. The electrochemical performance of the solid-state batteries is also competitive compared with other reports. This work can address several key issues of PVDF-based electrolytes and provide an encouraging path towards their large-scale production in the practical application. We sincerely look forward to the reviewer's approval for this revised manuscript.

To evaluate the influence of MSs on the electronic conductivity of the composite electrolyte, we assembled symmetric cells with stainless steel spaces as electrodes and conducted Chronoamperometry (CA) tests with an applied voltage of 500 mV (*Angew. Chem. Int. Ed.* 2023, e202217538; *Adv. Mater.* 2023, 35, 2208951). The experimental details are shown in lines 411-412, page 21. The results show that the electronic conductivity of PVDF, PVMS-10, PVMS-15, and PVMS-20 electrolytes is 6.82×10^{-11} , 2.08×10^{-10} , 3.09×10^{-10} , and 3.51×10^{-10} S cm⁻¹, respectively, as shown in Supplementary Fig. 23 and Table 2. The results indicate that the electronic conductivity is much smaller than the ionic conductivity (10^{-4} S cm⁻¹), although the electronic conductivity could be increased with the addition of MSs. When the electronic conductivity is sufficiently small, the conductivity corresponds to mainly ionic conductivity. Thus, we believe that the improved conductivity of the composite electrolyte is due to the ionic conductivity enhancement.

In addition, many semiconductors or conductors such as graphene (*Adv. Energy Mater.* 2022, 12, 2200967) and carbon particles (*Angew. Chem. Int. Ed.* 2023, e202217538)

were reported as fillers in the polymer-based composite electrolytes. The role of them can be mainly divided into three aspects: (1) the fillers could densify the local electric field distribution and help more efficient charge transfer to prevent local Li dendrite formation; (2) the enhanced bulk dielectric property helps to spared double layer region and salt solvation to enhance the interfacial and bulk charge transfer.

In all, we think the conductivity enhancement is mainly attributed to ionic conductivity in the composite PVMS-15 electrolyte.

Fig. 2d Relationship between the MSs content and β -PVDF ratio and ϵ_r' .

Supplementary Figure 23. Electronic conductivities tests.

Supplementary Table 2. Electronic conductivities of the electrolytes.

Electrolyte	Electronic conductivity ($S\text{ cm}^{-1}$)
PVDF	6.82×10^{-11}
PVMS-10	2.08×10^{-10}
PVMS-15	3.09×10^{-10}
PVMS-20	3.51×10^{-10}

As a final note here, it is unclear how much solvent (DMF) is present in the composite. Can the authors provide a weight or volume percentage in the experimental description of the sample preparation?

Reply: Thank you very much for your significant comment. The content of DMF solvent is determined by TGA and ss-NMR tests.

Thermogravimetric analysis (TGA) measurements show that the contents of the DMF solvent for PVMS-15 and PVDF electrolytes are approximately 12.46 wt% and 13.95 wt% (Supplementary Fig. 7), respectively, which are consistent with the solid-state nuclear magnetic resonance (ss-NMR) test results (Supplementary Fig. 8), indicating that the addition of MSs does not affect the solvent content. (line 123-127, page 6)

Supplementary Figure 7. Solvent content tested by TGA. TGA curves of PVDF (a) and PVMS-15 (b) electrolytes.

Supplementary Figure 8. Solvent content tested by ss-NMR. ^1H spectra of PVDF (a) and PVMS-15 (b) electrolyte.

Despite the limited knowledge on the transport processes taking place in the composite polymer electrolyte, incorporating such a material in cells seems to have some positive effect in the cycling performance (Figure 4). However, I think it is unfair to the readers to present such small plots, and also to have the axis for the Coulombic Efficiencies range from 0 to 100. At such a scale it is very difficult to ascertain what is the “real” behavior of the cells.

Reply: Thank you very much for your significant comment. To clearly show the positive effect in the cycling performance with the PVMS-15 electrolyte, we have revised the Fig. 4. It is seen that the coulombic efficiency of the batteries with PVMS-15 electrolyte is higher than that with the PVDF electrolyte, demonstrating the enhanced stability and suppressed side reactions.

Fig. 4 Electrochemical performance of Li||NCM811 full cells.

The comparison to other reports in Figure 4g is unreasonable because the authors do not consider difference of CAM loadings, which are known to have a strong effect on cycle number and rate capabilities in SSBs. Along these lines, I consider a CAM loading of ca. 2 mg/cm² not very competitive to other reports in SSBs. For example, sulfide- and halide- solid electrolyte based SSBs (press-cells) with CAM loadings of 10–20 mg/cm² CAM loadings have been reported by the Nazar and Janek groups recently. At the low loadings reported here, it is difficult to gauge the transferability of these results for higher loadings and/or thicker cathodes.

Reply: Thank you very much for your significant comment. Indeed, the electrochemical performance could be influenced by several factors, including the areal capacity, the current density, the ionic conductivity of electrolyte, the cut-off voltage, the thickness of electrolyte and so on. To show a fair comparison, we revised the Fig. 4g and added Supplementary Table 5. During the revision, we added the electrochemical performance of Li|PVMS-15|NCM811 cells with 8 mg cm⁻² (1.44 mAh cm⁻²) loading to evaluate the application potential of the PVMS-15 electrolyte (Fig. 4e). It should be noted that this loading is the highest loading compared with all reported solid-state batteries using the

PVDF-based electrolytes. However, the performance of the solid-state batteries using polymer-based electrolytes is still unsatisfactory compared with that using sulfide- and halide-based electrolytes due to their poor mechanical properties, which motivated us to further modify the polymer-based electrolytes.

Fig. 4g Comparison of the electrochemical performance of the device in this work with that of other recently reported solid-state batteries using PVDF-based and other polymer-based electrolytes.

Supplementary Table 5. Comparison of our developed batteries with other reported batteries.

Electrolyte	Ionic conductivity ($\times 10^{-4}$ S cm^{-1})	Areal capacity (mAh cm^{-2})	Current density (mA cm^{-2})	Thickness (μm)	Cathode material	Cut-off Voltage (V)	Cycle number	Ref.
PVDF/LLZTO	5	0.342	0.1	100	LCO	4.2	120	3
PVDF- b -PTFE/LLTO	1.38	0.45	0.225	61	NCM532	4.35	550	4
PVDF-HFP/LiTFSI	1.24	0.378	0.075	100	NCM532	4.3	200	5
PVDF-HFP/LiTFSI	2.7	0.36	0.36	—	NCM811	4.2	800	6
P(VDF-TrFE-CTFE)/LiTFSI	3.1	0.272	0.136	70	LFP	4.2	200	7

PVDF/BTO/LLTO	8	0.288	0.288	67	NCM811	4.3	1500	8
P(VDF-TrFE-CTFE)/(Pyr13-TFSI)	5.75	0.17	0.17	130	LFP	4.2	1000	9
PEO/SN/FEC	10.1	0.34	0.17	300~400	LFP	3.9	2000	10
IL/VEC/OFHDOD A /LiTFSI	13.7	0.187	0.099	100~200	NCM532	4.5	200	11
COF/LiClO ₄	1.2	0.17	0.17	80~100	LFP	4.2	750	12
COF/LiTFSI	1.65	0.425	0.1	32	LFP	3.8	130	13
PVMS-15	6.5	0.36	1.08	80	NCM811	4.3	2000	This work
PVMS-15	6.5	0.36	0.72	80	NCM811	4.5	600	This work
PVMS-15	6.5	0.36	0.72	80	NCM811	4.3	2000	This work
PVMS-15	6.5	1.44	0.288	80	NCM811	4.3	100	This work

Reviewer #3 (Remarks to the Author):

The authors report a PVDF-based gel electrolyte with MoSe₂ fillers. The MoSe₂ sheets are able to interact with PVDF molecules and alter the solvation structure of Li⁺. As a result, the electrolyte shows improved density, ionic conductivity, electrochemical stability, and mechanical properties. The authors have performed thorough characterizations and show decent electrochemical performances. I suggest that this article can be published after minor revisions. The following questions and advices are for the authors:

Reply: Thank you very much for the quite positive comments on our work. We have carefully revised our manuscript point by point according to your valuable comments

below.

1. Please label the molecules properly in Fig. 1b.

Reply: Thank you very much for your significant comment. We have labeled the molecules properly in Fig. 1b.

Fig. 1b Solvation structure control.

2. Please distinguish “interface” and “interphase”.

Reply: Thank you very much for your significant comment. We have revised incorrect terms and changed them to the solid electrolyte interphase (SEI) and cathode electrolyte interphase (CEI).

3. The SEM images of PVMS-15 and PVDF should be compared at the same magnification.

Reply: Thank you very much for your significant comment. We have revised the SEM images of PVDF and PVMS-15 electrolytes at the same magnification (Fig. 1e and f, Supplementary Fig. 3 and 4).

Fig. 1 Surface SEM images of the PVMS-15 (e) and PVDF (f) electrolyte.

Supplementary Figure 3. SEM images of the PVMS-15 electrolyte. a 500×. b 1500×. c 2000×.

Supplementary Figure 4. SEM images of the PVDF electrolyte. a 500×. b 1500×. c 2000×.

Supplementary Figure 6. Cross sectional SEM images of the electrolytes. a PVDF electrolyte. b PVMS-15 electrolyte and its EDS mappings of Mo (c) and F (d) element.

4. In Fig. 1e, it seems that the electrolyte is not homogeneous and there is a phase separation. The PVDF phase looks still a bit porous. Please provide more SEM images at different areas with higher magnifications.

Reply: Thank you very much for your significant comment. We have provided more optical photographs and SEM images of PVDF and PVMS-15 electrolytes at the same magnification. As shown in Supplementary Fig. 2, the optical photographs show that the MSs are uniformly distributed in the PVDF matrix to form a homogeneous film (6

cm × 6 cm). The SEM images show that PVDF electrolyte exhibits a porous structure due to the phase separation between polymer and solvent during the preparation process. In contrast, the PVDF spherulites present strong adhesion with MSs due to the interactions to form a dense and flat PVMS-15 electrolyte.

Fig. 1 Surface SEM images of the PVMS-15 (e) and PVDF (f) electrolyte.

Supplementary Figure 3. SEM images of the PVMS-15 electrolyte. a 500×. b 1500×. c 2000×.

Supplementary Figure 4. SEM images of the PVDF electrolyte. a 500×. b 1500×. c 2000×.

Supplementary Figure 6. Cross sectional SEM images of the electrolytes. a PVDF electrolyte. **b** PVMS-15 electrolyte and its EDS mappings of Mo (**c**) and F (**d**) element.

Supplementary Figure 2. Optical photographs of PVDF and PVMS-15 electrolytes. a PVDF electrolyte. **b** PVMS-15 electrolyte and rollability (**c**) and foldability (**d**) test.

5. 15% MoSe₂ in the polymer should be discontinuous in the electrolyte and tends to aggregate according to Fig. 1e. Do you really have a SEI with a uniform chemical

composition? The Li_2Se and Mo content in the SEI should vary dramatically at different spots.

Reply: Thank you very much for your quite significant comment. To evaluate the composition and distribution of the formed SEI by PVMS-15 electrolyte, we conducted the XPS depth (20 nm) profiling and time-of-flight secondary ion mass spectrometry (TOF-SIMS) measurements to evaluate the composition and distribution in a $200\ \mu\text{m} \times 200\ \mu\text{m} \times 50\ \text{nm}$ area and analyzed the results carefully.

As shown in Fig. 5a and Supplementary Fig. 44, the C and O atomic contents are decreased with the PVMS-15 electrolyte, demonstrating that DMF decomposition is effectively mitigated. The Se and Mo contents at different sputtering times remain constant at approximately 4.68% and 2.53%, respectively, indicating a uniform distribution of Li_2Se and Mo. We also performed time-of-flight secondary ion mass spectrometry (TOF-SIMS) measurements to evaluate the composition in a $200\ \mu\text{m} \times 200\ \mu\text{m} \times 50\ \text{nm}$ area. As shown in Fig. 5b and Supplementary Fig. 45, many organic species (CN^- , C^- and CNO^-) and inorganic species (Li_3OH^+ , Li_3CO_3^+ , Li_3O^+ , Li_3Se^+ , Mo^+ and F^-) can be obtained, which is consistent with the XPS results. In detail, the intensity profiles and corresponding three-dimensional (3D) views indicate that inorganic species are uniform and dominant in the SEI. (line 322-331, page 18)

To evaluate the stability of the SEI and its influence on Li deposition morphology, we assembled the Li||Cu cells. As shown in Supplementary Fig. 31, the Li|PVMS-15|Cu cell delivers the cycle life of 55 cycles with an average CE as high as 97.8%, while the Li|PVDF|Cu cell presents a short circuit after only 12 cycles with a low average CE of 86.4%. After plating $1\ \text{mAh cm}^{-2}$ Li on the Cu foil, dense and connected Li particles can be observed with the PVMS-15 electrolyte (Supplementary Fig. 32), while small and uneven Li particles are induced by the PVDF electrolyte (Supplementary Fig. 10). These results prove that the formation of Li_2Se in the SEI can significantly enhance the Li metal compatibility. (line 256-263, page 14)

Fig. 5 **a** Atomic ratios of the SEI at different sputter time with the PVMS-15 electrolyte. **b** TOF-SIMS depth profiles of the secondary ion fragments and 3D view (c) in the formed SEI by PVMS-15 electrolyte.

Supplementary Figure 45. TOF-SIMS depth profiles of secondary ion fragments in the formed SEI by PVMS-15 electrolyte.

Supplementary Figure 31. CE tests in Li||Cu cells using PVDF and PVMS-15 electrolytes.

Supplementary Figure 32. SEM images of the Li deposition obtained by plating 1 mAh cm⁻² Li on Cu substrate at 0.1 mA cm⁻² in Li||Cu cells using PVMS-15 electrolyte. a 500×. b 2000×.

Supplementary Figure 10. SEM images of the Li deposition obtained by plating 1 mAh cm⁻² Li on Cu substrate at 0.1 mA cm⁻² in Li||Cu cells using PVDF electrolyte. a 500×. b 2000×.

6. Please add a scale bar in Supplementary Fig. 31.

Reply: Thank you very much for your significant comment. We have added scale bars in this figure.

Supplementary Figure 46. Element EDS mappings for cycled Li metal anode using the PVMS-15 electrolyte.

7. The authors claim that the Li⁺-DMF coordination is strengthened in the electrolyte, which should also facilitate the decomposition of DMF. However, the authors claim the

opposite. How does the Li₂Se-containing SEI suppress DMF decomposition?

Reply: Thank you very much for your significant comment. The decomposition of DMF consists of chemical decomposition and electrochemical decomposition. For Li metal anode compatibility, the MoSe₂ can serve as interlayer between PVMS-15 electrolyte and Li metal to suppress the chemical decomposition of DMF. During cycling of the batteries, it is interesting that the formation potential of Li₂Se (1.5 V) is higher than that of the DMF reduction (~ 0.5 V) (*Nat. Commun.* 2020, 11, 4188; *Phys. Chem. Chem. phys.* 2020, 22, 8853-8863), which could also suppress side reactions and assist to enhance the electrochemical stability of the PVMS-15 electrolyte. For the NCM811 cathode compatibility, the dense structure of PVMS-15 electrolyte leads to uniform distribution of the DMF solvent, as evidenced by Fig. 1g, which can suppress the locally high-concentrated DMF decomposition, thus the electrochemical stability window of the PVMS-15 can be extended to 4.7 V, contributing to robust cycling under cut-off voltage of 4.5 V.

8. What is the ion-conductor in the NMC811 cathode?

Reply: Thank you very much for your significant comment. We are very sorry to provide insufficient experimental details. To construct ion transport channels inside the NCM811 cathode, we added Li salt (LiTFSI) during the preparation of NCM811 cathode as previous reports (*Batteries & Supercaps* 2020, 3, 876-883; *Adv. Mater.* 2019, 31, e1806082). We have added the details of experimental methods.

The cathode was prepared by mixing LiNi_{0.8}Co_{0.1}Mn_{0.1}O₂ (NCM811, D₅₀=3.77 μm), Super P (particle size: 40~50 nm, 99.5%), PVDF 5130 binder (M_w=1,200,000Da, particle size: 100 μm, 99.5%) and LiTFSI (Sigma-Aldrich, 99.95%) in a weight ratio of 75:10:10:5 (0.75 g: 0.1g: 0.1 g: 0.05 g) in N-methyl-2-pyrrolidone (NMP, 99.9%, Aladdin), followed by casting the slurry on an Al foil (thickness: 10 μm, 99.9%). (line 431-434, page 21)

9. Please specify the loading and/or areal capacity of the cathode and the thickness of the Li anode in the main text.

Reply: Thank you very much for your significant comment. We are very sorry to provide insufficient experimental details. For the coin cell, the cathode loading is 2 mg cm^{-2} and the thickness of Li foil is $450 \text{ }\mu\text{m}$. During the revision, we also added the electrochemical performance of Li|PVMS-15|NCM811 cells with 8 mg cm^{-2} loading (the highest loading compared with other reported batteries using the PVDF-based electrolytes) to evaluate the application potential of the PVMS-15 electrolyte (Fig. 4e). For the pouch cell, single side coated cathode (mass loading: 2 mg cm^{-2} , $4 \text{ cm} \times 4.5 \text{ cm}$) and one double coated anode ($4.5 \text{ cm} \times 5 \text{ cm}$, $20 \text{ }\mu\text{m}$ Li for each side) were stacked one by one and separated by PVMS-15 electrolyte ($5 \text{ cm} \times 5.5 \text{ cm}$). We have added the loading of the cathode and the thickness of the Li anode in the main text.

Fig. 4e Cycling stability of Li|PVMS-15|NCM811 cells with 8 mg cm^{-2} .

10. Fig. 4g is not a fair comparison. Please add an axis of areal capacity and change the C-rate to current density in the graph.

Reply: Thank you very much for your significant comment. Indeed, the electrochemical performance could be influenced by several factors, including the areal capacity, the current density, the ionic conductivity of electrolyte, the cut-off voltage, the thickness of electrolyte and so on. To show a fair comparison, we revised the Fig. 4g and added Supplementary Table 5.

Fig. 4g Comparison of the electrochemical performance of the device in this work with that of other recently reported solid-state batteries using PVDF-based and other polymer-based electrolytes.

Supplementary Table 5. Comparison of our developed batteries with other reported batteries.

Electrolyte	Ionic conductivity ($\times 10^{-4}$ S cm^{-1})	Areal capacity (mAh cm^{-2})	Current density (mA cm^{-2})	Thickness (μm)	Cathode material	Cut-off Voltage (V)	Cycle number	Ref.
PVDF/LLZTO	5	0.342	0.1	100	LCO	4.2	120	3
PVDF- b -PTFE/LLTO	1.38	0.45	0.225	61	NCM532	4.35	550	4
PVDF-HFP/LiTFSI	1.24	0.378	0.075	100	NCM532	4.3	200	5
PVDF-HFP/LiTFSI	2.7	0.36	0.36	—	NCM811	4.2	800	6
P(VDF-TrFE-CTFE)/LiTFSI	3.1	0.272	0.136	70	LFP	4.2	200	7
PVDF/BTO/LLTO	8	0.288	0.288	67	NCM811	4.3	1500	8

P(VDF-TrFE-CTFE)/(Pyr13-TFSI)	5.75	0.17	0.17	130	LFP	4.2	1000	9
PEO/SN/FEC	10.1	0.34	0.17	300~400	LFP	3.9	2000	10
IL/VEC/OFHDOD A /LiTFSI	13.7	0.187	0.099	100~200	NCM532	4.5	200	11
COF/LiClO ₄	1.2	0.17	0.17	80~100	LFP	4.2	750	12
COF/LiTFSI	1.65	0.425	0.1	32	LFP	3.8	130	13
PVMS-15	6.5	0.36	1.08	80	NCM811	4.3	2000	This work
PVMS-15	6.5	0.36	0.72	80	NCM811	4.5	600	This work
PVMS-15	6.5	0.36	0.72	80	NCM811	4.3	2000	This work
PVMS-15	6.5	1.44	0.288	80	NCM811	4.3	100	This work

REVIEWER COMMENTS

Reviewer #1 (Remarks to the Author):

I thank the authors for their response, most of which is quite satisfactory and I think the manuscript is much improved. However, there remain a couple of things that I would like clarified, and I am not yet able to recommend publication:

1. The authors have added Supplementary Figures 11 and 30, both of which show dendrite growth at 0.1 mA cm⁻², all be it after different capacities. Growth of Li dendrites at this current density is highly concerning as it appears to undermine the authors' claims to high critical currents elsewhere in the manuscript, suggesting that this is only possible due to the impractically low capacities used in cycling.
2. XPS fitting of PVMS-15 shows that metallic Mo, which will be electronically conductive, is formed in the interphase. This can lead to unstable interphase growth with Li metal, which does not self-passivate. Can the authors comment on why their interphase appears to be stable despite there being metallic Mo in the interphase?

As a more minor point, the XPS fittings, Supplementary Figures 43 and 48, need to be larger and higher resolution. As they are, it is not possible to assess the fitting.

As requested, I have prepared a brief comment on the authors' response to reviewer 2:

The authors satisfactorily address the main comments of reviewer 2 on electronic conductivity and solvent content. However, I feel that the potential for MoSe₂ to conduct electronically makes this electrolyte non-viable for real battery systems (even though the overall electronic conductivity is low), and limits its interest to a theoretical understanding of how addition of secondary phases can improve performance in this class of polymer electrolytes. While interesting, I am not convinced that this is of high level of interest to a wide readership.

Reviewer #2 (Remarks to the Author):

The authors addressed most of my comments, and the quality of the manuscript is improved. However, the authors claimed a NMC811 content of 80% in the cathode in the initial version, but they claim a different content of 75% to reply to my comment. I find it inconsistent and dishonest.

Please explain why the addition of LiTFSI in the cathode can provide sufficient Li⁺ conductivity. Is there also solvent or liquid electrolyte in the cathode?

Response to Referees' Comments

Dear Editors and Referees:

Thank you very much again for your time in reviewing our manuscript (Manuscript ID: NCOMMS-22-53461). We appreciate all the reviewers for their important and insightful comments and suggestions, which is significant for us to further raise the quality of our work. We have carefully read the comments and revised the manuscript accordingly. All comments have been replied point-by-point as below and the revised parts have been highlighted in **BLUE** in the revised manuscript. We sincerely hope this could take away the reviewer's concerns and expect this revised manuscript can reach the high standard of *Nature Communications*.

Referees' comments:

Reviewer #1 (Remarks to the Author):

I thank the authors for their response, most of which is quite satisfactory and I think the manuscript is much improved. However, there remain a couple of things that I would like clarified, and I am not yet able to recommend publication:

Reply: Thank you very much for your quite positive impression on our first revision. We also sincerely appreciate the reviewer's feedback for us to further improve the quality of our work. We feel that the reviewer's significant comments are worth to be discussed, which inspired us to perform many additional experiments, including further exploring real batteries' performance under more extreme conditions (coin cells: high-loading 2.6 mAh cm⁻² NCM811 cathode; pouch cells: 1.44 mAh cm⁻² NCM811 cathode, 50 μm Li anode, negative to positive areal capacity (N/P) ratio = 6.25, 15 mAh capacity in total), understanding the ion transport inside the solid-state NCM811 cathode and

revealing the electronic conductivity of the formed solid electrolyte interphase (SEI). Based on the obtained results, we have carefully revised the manuscript and provided intensive discussions.

1. The authors have added Supplementary Figures 11 and 30, both of which show dendrite growth at 0.1 mA cm^{-2} , all be it after different capacities. Growth of Li dendrites at this current density is highly concerning as it appears to undermine the authors' claims to high critical currents elsewhere in the manuscript, suggesting that this is only possible due to the impractically low capacities used in cycling.

Reply: Thank you very much for this significant comment. We surmise the reviewer has doubts about the result of critical current density (CCD) and critical deposition capacity (CDC). In response to this point, we have systematically discussed it as below. At first, we would like to analyze their definitions and the related meanings. The critical current density (CCD) is defined as the maximum endurable current density of Li metal battery before failure, which can identify the rate-determining steps of Li kinetics in a battery. The critical deposition capacity (CDC) is defined as the maximum endurable capacity of Li metal anode during cycling before Li dendrite growth and battery circuit short, which can reflect the homogeneity of Li plating on Li metal anode and indicate the compatibility with high-loading cathode in a battery system. With this in mind, the CCD and CDC with different meanings are complementary with each other to determine the stability and performance of Li metal anode. **Therefore, both the CCD and CDC results are consistent with our design logic. This could also support our conclusion that dense PVDF-based electrolyte is quite important to endow fast ion transport kinetics, uniform Li deposition on the Li metal anode and then boost the performance of Li||NCM811 solid-state full batteries towards practical application.** Inspired by the reviewer's insightful comments, we further analyzed the CCD and CDC results and evaluated the performance of Li||NCM811 full batteries using our developed PVMS-15 electrolyte under more practical conditions and obtained encouraging results.

At present, we have revised the manuscript accordingly. We hope this could take away the reviewer's concerns.

We would like to further explain the above reply into details as below:

(1) Our intention and analysis of the CCD results. As mentioned in the manuscript, the uncontrollable side reactions of the residual DMF with Li metal lead to continuous decomposition of the electrolyte and unsatisfactory capacity retention of the batteries. In previous report [*Batteries & Supercaps* **3**, 876-883 (2020)], it has been identified that the **side reactions could be obviously accelerated under high current density**, which severely hinders the battery's operation at a high current density (usually $\geq 0.5 \text{ mA cm}^{-2}$ for Li||Li cells and full cells). Given this, it is reasonable to test the CCD to evaluate the (electro)chemical stability and Li^+ transport kinetics of our developed PVMS-15 electrolyte. Considering the CCD result could be influenced by many parameters and there lacks a standard protocol, we adopted Time control and Capacity control methods [*Adv. Funct. Mater.* **31**, 2009925 (2021)], as shown in Fig. 3n and Supplementary Fig. 28, respectively. For the Time control method, the Li||Li symmetric cells were cycled for 5 times at current density of 0.05 mA cm^{-2} to form the solid electrolyte interphase (SEI), which were then cycled from 0.1 mA cm^{-2} with the current density increasing 0.1 mA cm^{-2} per cycle, while the charge/discharge time remains fixed for 1 h. For the Capacity control method, the Li||Li cells were cycled for 5 times at current density of 0.05 mA cm^{-2} to form the SEI, which were then cycled from 0.1 mA cm^{-2} with the current density increasing 0.1 mA cm^{-2} per cycle, while the charge/discharge capacity remains fixed for 0.2 mAh cm^{-2} (a common value).

Consistent with our design logic, the CCD in the PVMS-15 electrolyte is much higher than that in the PVDF electrolyte, indicating its superior stability and suppressed side reactions. This is attributed to three reasons: 1) The MoSe_2 filler can serve as interlayer between PVMS-15 electrolyte and Li metal to **suppress the chemical decomposition** of DMF; 2) The cyclic voltammetry (CV) curves of Li||Cu cells (Supplementary Fig.

27) show that the formation potential of the Li₂Se (1.5 V) is higher than that of the DMF reduction (~ 0.5 V) [*Nat. Commun.* **11**, 4188 (2020); *Phys. Chem. Chem. Phys.* **22**, 8853-8863 (2020)], which could **suppress the electrochemical decomposition** of the DMF; 3) The PVDF electrolyte with porous structure could result in uneven electric field and poor interfacial contact [*Nat. Nanotechnol.* **18**, 602-610 (2023)], while the dense and flat structure of the PVMS-15 electrolyte helps to **form even electric field** and compact contact with Li metal, thereby effectively mitigating the “tip effect” and induce a uniform Li deposition to suppress Li dendrite growth at high current density. In addition, the polarization voltage in the PVMS-15 electrolyte is smaller than that in the PVDF electrolyte under same current density, due to the faster Li⁺ transport kinetics. **In summary, the CCD results prove that the PVMS-15 electrolyte can suppress the DMF decomposition and maintain stable at high current density, which therefore could support the batteries’ cycle at high current density/rate, as evidenced by the excellent cycling performance of Li||NCM811 full cells and Li||Li symmetric cells.**

Fig. 3n CCD test curves with Time control method.

Supplementary Figure 28. CCD test curves with Capacity control method.

(2) Our intention and analysis of the CDC results. Before the analysis, it is essential to discuss why the PVDF electrolyte induces uneven Li deposition. As we mentioned in the manuscript, the PVDF electrolyte processes a porous structure (Fig. 1f) and the DMF solvent aggregates around the PVDF spherulites (Fig. 1i), thus the residual DMF distribution is non-uniform, which results in that the de-solvation and reaction process of the $[\text{Li}(\text{DMF})_x]^+$ on the surface of Li metal is inhomogeneous. Consequently, uneven Li deposition and the “tip effect” could be observed. With the capacity increases, Li dendrite growths and the batteries occur short circuit. The uneven Li deposition (Supplementary Fig. 10) and the Li dendrites growth (Fig. 5d) have been proved in the manuscript. While in case of the PVMS-15 electrolyte, the dense structure (Fig. 1e) enables uniform DMF solvent distribution (Fig. 1h), thereby realizing dense and flat Li deposition morphology (Supplementary Fig. 32). **Herein, to further quantitatively evaluate the effect of dense electrolyte structure on the uniform Li deposition, we tested the CDC.** We assembled the Li||Li cells using the PVDF and PVMS-15 electrolytes. The cells were cycled 5 times under current density of 0.1 mA cm^{-2} to form the SEI, which were then operated by continuously charge (one side Li metal stripping and the other side Li metal plating) until the battery presents a short circuit.

As shown in Supplementary Fig. 11, the CDC in the PVDF electrolyte is approximately 1.0 mAh cm^{-2} . In contrast, the CDC in the PVMS-15 electrolyte could achieve 2.6 mAh cm^{-2} (Supplementary Fig. 30). This result further confirmed that dense structure of the PVMS-15 electrolyte is powerful to achieve uniform Li deposition. **More importantly, it tells us that the PVMS-15 electrolyte is compatible with a high-loading cathode in the Li||NCM811 full cell.** To confirm this point, we tried to assemble the coin cells using 2.6 mAh cm^{-2} NCM811 cathode, PVMS-15 electrolyte and Li metal anode. As shown in Supplementary Fig. 39, the cells can stably cycle without short circuit at 0.1C (0.26 mA cm^{-2}). It should be noted that this cycling test is still underway and is expected to last for prolonged cycles. **On this basis, we could respectfully claim that the CDC result is acceptable, which is not only consistent with our design proposal and**

conclusion of this work, but also highlights the reliability of the developed PVMS-15 electrolyte towards practical application. Beyond, it is worth noting that this performance is quite competitive among the currently reported results using the PVDF-based and other polymer-/composite-based electrolyte systems, as shown in Fig. R1.

In summary, we sincerely appreciate the reviewer's this comment. After careful analysis and discussion, we believe the CCD and CDC results are acceptable and consistent with our design logic and conclusion. We have revised the manuscript accordingly. The authors hope this could take away the reviewer's concerns.

Supplementary Figure 11. CDC test curves in the PVDF electrolyte.

Supplementary Figure 30. CDC test curves in the PVMS-15 electrolyte.

Supplementary Figure 39. Electrochemical performance of the Li||NCM811 coin cells with 2.6 mAh cm⁻² cathode at 0.1C. a Cycling stability. **b** Charge/discharge curve.

Fig. R1 Areal capacity comparison of this work with other reported results. The detailed comparison of the performance is listed in Supplementary Table 6.

2. XPS fitting of PVMS-15 shows that metallic Mo, which will be electronically conductive, is formed in the interphase. This can lead to unstable interphase growth with Li metal, which does not self-passivate. Can the authors comment on why their interphase appears to be stable despite there being metallic Mo in the interphase?

Reply: Thank you very much for this significant comment. We appreciate reviewer's query on the properties of SEI, which indeed should be carefully discussed. Inspired by

the insightful comment of the reviewer, we measured the electronic conductivities of the formed SEI and explained why the SEI is stable as below:

(1) To evaluate the electronic conductivities of the formed SEI, we directly measured them following our previous reported method [*Nano Lett.* **20**, 6606-6613 (2020)]. Since the SEI has similar properties with the solid-state electrolyte (SSE), the SEI could be considered as a thin SSE layer [*Nat. Nanotechnol.* **14**, 1042-1047 (2019)]. Therefore, the assembly of the device for investigating the electronic conductivity of the SEI can follow the Hebb-Wagner method for measuring the electronic conductivity of the SSE. Given this, we cycled the Li||NCM811 full cells for 20 times at 0.5C and 25°C and then disassembled them to obtain the SEI-coated Li metal anode (donated as SEI-Li). The SEI-Li was clamped by two steel plates, and then the (-) Li/SEI/blocking electrode (+) cells were assembled. A dc voltage was applied in a chronoamperometric mode to the cells using an AMETEK. In the stationary state, the remaining current only stems from electrons diffusing through the SEI [*J. Chem. Phys.* **20**, 185-190 (1952); *J. Electrochem. Soc.* **143**, 2198-2203 (1996)]. The resistance of the SEI (R_e) could be calculated by:

$$R_e = \frac{\Delta V}{\Delta I_t}$$

where ΔV is the step voltage of 10 mV, and ΔI_t is the steady state response current.

Thus, the electronic resistivity of the SEI (ρ) could be calculated by:

$$\rho = \frac{R_e \times A}{L}$$

where A is the area of SEI (2 cm^2), L is the thickness of SEI which has been obtained by cryogenic scanning transmission electron microscopy (cryo-STEM). As shown in Supplementary Fig. 49, **the electronic resistivity of the SEI in PVMS-15 electrolyte is $4.23 \times 10^5 \text{ } \Omega \text{ cm}$, almost equal to that in the PVDF electrolyte ($4.21 \times 10^5 \text{ } \Omega \text{ cm}$). The electronic conductivity is calculated to be $2.36 \times 10^{-6} \text{ S cm}^{-1}$ and $2.38 \times 10^{-6} \text{ S cm}^{-1}$, respectively. This result strongly demonstrates that the formation of metallic Mo and Mo-containing species show negligible influence on the electroconductivity of the SEI.** In addition, the value of the electronic conductivities is acceptable for practical lithium metal batteries [*Nat. Energy* **2**, 17119 (2017)].

Supplementary Figure 49. Current-time curves of the Li/SEI/SS cells for the electronic conductivity of the SEI in PVMS-15 (a) and PVDF (b) electrolytes.

(2) We then explained this phenomenon based on our understandings. 1) According to the X-ray photoelectron spectroscopy (XPS) results for the SEI, the ratio of Mo element is only 2.5%, **which can hardly form a continuous network for electron transport.**

2) We have directly characterized the SEI via cryo-STEM, but metallic Mo and Mo-containing species were not observed, which is much likely due to that **these species are surround by the organic component inside the SEI**, indicating its quite limited role to transport electrons.

(3) Many famous works have also supported us to deepen our fundamental understandings of the SEI. Indeed, an ideal SEI on Li metal should delivers highly ionically conductive, electronically insulative, chemically stable against Li, and capable of separating the direct contact between Li and electrolyte [*Adv. Energy Mater.* **11**, 2003092 (2021); *Nat. Commun.* **13**, 5431 (2022)]. However, it was proved that when the electron conductive phase is coated with electron insulating phase in the SEI, the risk of electron transport could be neglected. For instance, Prof. Nazar's group reported various electron conductive alloys (Li_yM_z , $\text{M}=\text{In, Zn, Bi, As}$) to protect Li metal anode [*Nat. Energy* **2**, 17119 (2017)]. When the insulating LiCl was employed on the surface of these alloys, the protective layer could effectively prevent the electron transport. With this in mind, it is reasonable to note that trace amount of metallic Mo and Mo-containing species in the SEI can hardly transport electrons.

(4) The instability of the DMF with Li metal is the biggest challenge for PVDF-based electrolyte [*Batteries & Supercaps* **3**, 876-883 (2020)]. To further explain why the

formed SEI in PVMS-15 electrolyte is stable, we here provide detailed analysis. 1) The prior formation of Li_2Se component effectively suppresses the DMF reduction, which therefore reduces the amount of C-containing organic phase and LiOH and Li_2CO_3 , as evidenced by the XPS results. This contributes to enhanced stability of the SEI with Li metal [*Angew. Chem. Int. Ed.* **60**, 11442-11447 (2021); *Nat. Commun.* **13**, 2575 (2022)], as proven by higher coulombic efficiency (Supplementary Fig. 31) of the Li||Cu cells. 2) The modified SEI composition possesses higher Young's modulus of 4.73 GPa (Fig. 5f), which can significantly enhance the robustness and interfacial energy of the SEI, thus blocking Li dendrites growth [*Nat. Commun.* **12**, 5746 (2021)]. 3) The dense structure of the PVMS-15 electrolyte helps to form dense SEI (Fig. 5e), which prevents the direct contact of DMF with Li metal and mitigates continuous consumption of DMF during cycling. 4) Li_2Se with a quite low ion diffusion barrier (only 0.056 eV) could significantly enhance the ion transport capability of the SEI (Fig. 3i). **With the above-mentioned merits, the SEI in the PVMS-15 electrolyte with enhanced stability and interfacial kinetics contributes to robust cycling of the batteries.**

As a more minor point, the XPS fittings, Supplementary Figures 43 and 48, need to be larger and higher resolution. As they are, it is not possible to assess the fitting.

Reply: Thank you much for this significant comment. We have revised the figures in the revised manuscript.

Supplementary Figure 44. XPS results of SEI using PVDF and PVMS-15 electrolytes.

Supplementary Figure 50. XPS results of CEI using PVDF and PVMS-15 electrolytes.

As requested, I have prepared a brief comment on the authors' response to reviewer 2: The authors satisfactorily address the main comments of reviewer 2 on electronic

conductivity and solvent content. However, I feel that the potential for MoSe₂ to conduct electronically makes this electrolyte non-viable for real battery systems (even though the overall electronic conductivity is low), and limits its interest to a theoretical understanding of how addition of secondary phases can improve performance in this class of polymer electrolytes. While interesting, I am not convinced that this is of high level of interest to a wide readership.

Reply: Thank you much for this valuable comment. In terms of electronic conductivity of the electrolyte, we compared the PVMS-15 electrolyte with other reported systems. Inspired by the constructive suggestions, we also tried to assemble the lab-level single-layer pouch cells (size: 3 cm × 3.5 cm; total cathode active material loading: 84 mg; capacity: 15 mAh; N/P = 6.25) to further evaluate the performance of Li||NCM811 full cells under real/extreme conditions and obtained encouraging result.

(1) At first, we added Supplementary Table 3 to compare the electronic conductivity (σ_e) of our developed PVMS-15 electrolyte with other reported electrolytes, including sulfide-, halide-, oxide-, chloride-, polymer- and composite-based electrolytes. We would like to respectfully point out that the σ_e of our PVMS-15 electrolyte is as low as enough among the various systems to meet the requirements of practical batteries.

Supplementary Table 3. Comparison of electronic conductivities of the PVMS-15 electrolyte with other reported electrolyte systems.

Electrolyte	Electronic conductivity (S cm ⁻¹)	Reference
Li ₂ In _{1/3} Sc _{1/3} Cl ₄	4.7×10 ⁻¹⁰	Nat. Energy 7 , 83-93 (2022)
Li ₃ YCl ₆	2.8×10 ⁻⁹	Adv. Mater. 30 , 1803075 (2018)
PEO/PVDF/LiTFSI/TiO ₂	10 ⁻¹⁰	Adv. Sci. 9 , 2200213 (2022)
LiTFSI/EmimFSI/PMMA/LLZO	3.14×10 ⁻¹⁰	Adv. Mater. 34 , 2205560 (2022)
Li ₃ PS ₄	5×10 ⁻⁹	Energy Environ. Sci. 11 , 2828-2832 (2018)
LiF-Li ₁₀ GeP ₂ S ₁₂	2.42×10 ⁻⁹	Adv. Mater. 35 , 2211047 (2023)

LiNbOCl ₄	5.22×10 ⁻¹⁰	Angew. Chem. Int. Ed. 135 , e202217581 (2023)
PEO/LiTFSI/GDC	5.5×10 ⁻¹⁰	Angew. Chem. Int. Ed. 59 , 4131 (2020)
LLZO/Li ₃ AlF ₆	1.27×10 ⁻⁸	Adv. Mater. 35 , 2208951 (2023)
PI/LLZTO/PVDF	3.73×10 ⁻¹⁰	Energy Storage Mater. 26 , 283-289 (2020)
PEO/LiTFSI/carbon	6.83×10 ⁻⁹	Angew. Chem. Int. Ed. 62 , e202217538 (2023)
PVMS-15	3.09×10⁻¹⁰	This work

(2) The optical photograph and cycling stability of the Li||NCM811 solid-state pouch cell is shown in Supplementary Fig. 42. Much beyond our expectations, the pouch cell can normally cycle at 0.1C without addition of any other liquid electrolyte/solvent and is expected to achieve prolonged cycles. To the best of our knowledge, this is the first demonstration of the pouch cells with high capacity using the PVDF-based electrolytes. Therefore, our developed devices show excellent transferability towards the practical application.

Supplementary Figure 42. Cycling stability of the Li||NCM811 solid-state pouch cells.

(3) In terms of the interest of this work, we believe it is of both theoretical and practical significance after careful revisions as the following reasons.

1) In recent years, the PVDF-based electrolyte gets more and more attention and is expected as a promising candidate for high-performance solid-state lithium batteries [*Nat. Nanotechnol.* **17**, 768-776 (2023)]. However, its porous structure and the limited

stability bring safety risks. Addressing these issues is critical for practical application. Herein, for the first time, we found a simple and innovative approach of phase regulation to obtain dense PVDF-based electrolyte. More importantly, many associated benefits including improve the ionic conductivity, widen the electrochemical stability window of the electrolyte, suppress the side reactions with electrodes and increase the coulombic efficiency could be achieved by this ingenious design. We believe our findings and results could deepen the fundamental understandings of the PVDF-based electrolyte for the broad audience in this field.

2) Achieving high energy density and long cycle life have been considered as major challenges for solid-state lithium metal batteries. We firmly believe that **this is the best demonstration of PVDF-based electrolyte that simultaneously achieves a wide set of demanding properties under practical conditions**, including long cycle life (3000 times), high cut-off voltage (4.5 V), high current density (3C), wide temperature range (-20 ~ 45 °C), high cathode loading (2.6 mAh cm⁻²), and pouch cell level (15 mAh capacity) in this field. The performance of our developed devices is quite competitive, even beyond the PVDF-based solid-state batteries. This work provides an encouraging path towards their large-scale production in the practical application.

In all, we look forward to the reviewer's approval and hope this revised manuscript could reach the high standard of *Nature Communications*.

Reviewer #2 (Remarks to the Author):

The authors addressed most of my comments, and the quality of the manuscript is improved. However, the authors claimed a NMC811 content of 80% in the cathode in the initial version, but they claim a different content of 75% to reply to my comment. I find it inconsistent and dishonest.

Reply: Thank you very much for your quite positive comment of our first revision. We are very sorry to make mistakes for the experimental methods in the initial manuscript,

which makes it confused and suspicious for the reviewer to our work. In fact, we indeed added some Li salt (LiTFSI) when preparing the NCM811 cathode, in which the weight ratio of the NCM811:PVDF5130:Super P:LiTFSI is 75:10:10:5. This is referred to previous publications [*Adv. Mater.* 2019, **31**, 1806082; *Batteries & Supercaps* **3**, 876-883 (2020); *ACS Appl. Mater. Interfaces* **12**, 24837-24844 (2020)] from Prof. Ce-Wen Nan's group, who has made great contributions to the development of the PVDF-based electrolyte. However, the role of Li salt remains unclear and has always been neglected. Thus, we feel the reviewer's comment is insightful and important. Actually, we also paid attention to the ion transport inside cathode and we are conducting another work to investigate the role of added Li salt inside the cathode. In response to the reviewer's comment, we provided our discussions and perspectives based on our obtained results at present.

Please explain why the addition of LiTFSI in the cathode can provide sufficient Li⁺ conductivity. Is there also solvent or liquid electrolyte in the cathode?

Reply: Thank you very much for your quite insightful and constructive comment. We here provided our obtained results and findings on the investigation of the role of Li salt inside cathode, which will be systematically discussed in another work.

(1) First, we investigated whether residual NMP solvent exists or not inside the fresh NCM811 cathode via high-resolution nuclear magnetic resonance (NMR) technologies. We prepared two NCM811 cathodes: 1) Weight ratio of NCM811:PVDF5130:Super P:LiTFSI is 75:10:10:5; 2) Weight ratio of NCM811:PVDF5130:Super P is 80:10:10. The results are shown in Fig. R2. **It is revealed that there is no residual NMP solvent inside these cathodes.**

Fig. R2 ^1H NMR spectra of NCM811 cathode with LiTFSI (a) and without LiTFSI (b).

(2) Second, we found an interesting phenomenon that the solvent inside the electrolyte could spontaneously diffuse into the NCM811 cathode, and the addition of LiTFSI inside cathode could further promote the solvent diffusion. We assembled Li|PVMS-15|NCM811 (with LiTFSI) and Li|PVMS-15|NCM811 (without LiTFSI) cells and stored them for 5 h (Before the cell tests for cycling, we always stored the cells for 5 h to promote interfacial contact). Then we disassembled the cells and detected the solvent via Fourier transform infrared (FTIR) spectra. As shown in Fig. R3, obvious signal of the C=O group (1654 cm^{-1}) could be observed, demonstrating the DMF solvent in the PVMS-15 electrolyte could diffuse into the NCM811 cathode. In case of the cathode with LiTFSI, the stronger intensity of the C=O group indicates the enhanced DMF diffusion, which may be ascribed to the strong binding interactions/affinity of the LiTFSI with solvent. We would like to further investigate this finding.

Fig. R3 FTIR spectra of the NCM811 cathode.

(3) Third, we revealed the effect of LiTFSI on the ion transport behavior. We conducted the galvanostatic intermittent titration technique (GITT) measurements to investigate

its effect on Li^+ diffusion kinetics, as shown in Fig. R4. The diffusion coefficient is calculated to be $2.3 \times 10^{-10} \text{ cm}^2 \text{ s}^{-1}$ with the LiTFSI, higher than that without the LiTFSI ($1.1 \times 10^{-10} \text{ cm}^2 \text{ s}^{-1}$). This result indicates that the addition of LiTFSI could promote the solvent diffusion, thereby enhancing the ion transport kinetics inside NCM811 cathode.

Fig. R4 Li^+ diffusion coefficients tests of the NCM811 cathode. **a-b**, CV curves of the Li||NCM811 cells with LiTFSI (**a**) and without LiTFSI (**b**). **c**, Plots of peak current (I_p , derived from CV curves) as a function of the square root of the scan rate ($v^{1/2}$).

(4) Forth, we evaluated the rate capabilities of the Li|PVMS-15|NCM811 (with LiTFSI) and Li|PVMS-15|NCM811 (without LiTFSI) cells. As shown in Fig. R5, the discharge capacities paired with the NCM811 (with LiTFSI) cathode are much higher. **These results prove that the addition of LiTFSI inside the NCM811 cathode during preparation is useful to increase the capacity output.**

Fig. R5 Rate capabilities of the Li||NCM811 batteries with LiTFSI (**a**) and without LiTFSI (**b**) inside cathode.

In all, the addition of LiTFSI inside the NCM811 cathode is useful to enhance the ion transport kinetics and increase the capacity output. These findings are interesting and need further exploring. It is worth noting that the results can't affect the integrality and

accuracy of our work, in which we only modified the electrolyte while all the batteries are tested using the same cathode and anode. We have carefully checked the methods. We hope this could take away the reviewer's concerns and queries.

REVIEWERS' COMMENTS

Reviewer #1 (Remarks to the Author):

The authors have addressed my concerns to my satisfaction. I now feel that the manuscript is suitable for publication in Nature Communications.

Reviewer #3 (Remarks to the Author):

The authors have addressed all my comments, and the manuscript has significantly improved. I recommend accepting the manuscript with no further changes.

Response to Referees' Comments

Dear Editors and Referees:

Thank you very much for your patience and great contribution in reviewing our manuscript (Manuscript ID: NCOMMS-22-53461B). We sincerely appreciate the reviewers' approval and the editor's decision of our work for publication on *Nature Communications*.

Referees' comments:

Reviewer #1 (Remarks to the Author):

The authors have addressed my concerns to my satisfaction. I now feel that the manuscript is suitable for publication in Nature Communications.

Reply: We sincerely appreciate the reviewer's endorsement. All the comments and suggestions are valuable to raise the quality and novelty of our work.

Reviewer #3 (Remarks to the Author):

The authors have addressed all my comments, and the manuscript has significantly improved. I recommend accepting the manuscript with no further changes.

Reply: Thank you very much for your approval. We sincerely appreciate the reviewer's comments and suggestions to improve the significance of our work.